# A meet-up of two second messengers: the c-di-AMP receptor DarB controls (p)ppGpp synthesis in *Bacillus subtilis*

Larissa Krüger[1], Christina Herzberg[1], Dennis Wicke[1], Heike Bähre[2], Jana L. Heidemann[3], Achim Dickmanns[3], Kerstin Schmitt[4], Ralf Ficner [3] & Jörg Stülke [1✉]

Many bacteria use cyclic di-AMP as a second messenger to control potassium and osmotic homeostasis. In *Bacillus subtilis*, several c-di-AMP binding proteins and RNA molecules have been identified. Most of these targets play a role in controlling potassium uptake and export. In addition, c-di-AMP binds to two conserved target proteins of unknown function, DarA and DarB, that exclusively consist of the c-di-AMP binding domain. Here, we investigate the function of the c-di-AMP-binding protein DarB in *B. subtilis*, which consists of two cystathionine-beta synthase (CBS) domains. We use an unbiased search for DarB interaction partners and identify the (p)ppGpp synthetase/hydrolase Rel as a major interaction partner of DarB. (p)ppGpp is another second messenger that is formed upon amino acid starvation and under other stress conditions to stop translation and active metabolism. The interaction between DarB and Rel only takes place if the bacteria grow at very low potassium concentrations and intracellular levels of c-di-AMP are low. We show that c-di-AMP inhibits the binding of DarB to Rel and the DarB–Rel interaction results in the Rel-dependent accumulation of pppGpp. These results link potassium and c-di-AMP signaling to the stringent response and thus to the global control of cellular physiology.

[1] Department of General Microbiology, Institute for Microbiology & Genetics, GZMB, Georg-August-University Göttingen, Göttingen, Germany. [2] Research Core Unit Metabolomics, Hannover Medical School, Hannover, Germany. [3] Department of Molecular Structural Biology, Institute for Microbiology & Genetics, GZMB, Georg-August-University Göttingen, Göttingen, Germany. [4] Department of Molecular Microbiology and Genetics, Service Unit LCMS Protein Analytics, Institute for Microbiology & Genetics, GZMB, Georg-August-University Göttingen, Göttingen, Germany. ✉email: jstuelk@gwdg.de

All living cells contain high concentrations of potassium ions[1,2]. This ion is required for the activity of many enzymes and protein complexes, among them the ribosome, for buffering the negative charge of the DNA and for osmoadaptation[1,3]. On the other hand, potassium may become toxic if the intracellular concentration becomes too high[1]. Therefore, potassium homeostasis has to be carefully controlled. In many bacteria, a second messenger—cyclic di-AMP (c-di-AMP)—is involved in the control of potassium homeostasis[4,5]. The nucleotide is synthesized at high potassium concentrations, whereas low c-di-AMP levels indicate a potassium limitation[6]. The control of the homeostasis of potassium and other osmolytes is the reason that c-di-AMP is essential for many of the bacteria that produce this signaling nucleotide[7]. c-di-AMP acts by binding to a variety of targets to control their activity[7,8]. Among the targets of c-di-AMP are several proteins, like the potassium importers and exporters and a two-component sensor kinase as well as a riboswitch that are involved in the control of potassium homeostasis. Of all known second messenger nucleotides c-di-AMP is unique in binding and controlling both a protein and the mRNA molecule that encodes it. This is the case for the *Bacillus subtilis* KtrA and KimA potassium transporters that are both bound and thus inhibited by c-di-AMP. In addition, the corresponding mRNAs each carry a c-di-AMP responsive riboswitch, and binding of c-di-AMP prevents the expression of the transporters[4,6,9].

In *B. subtilis* and the related pathogen *Listeria monocytogenes*, the analysis of c-di-AMP-binding proteins identified two potential signal transduction proteins of unknown function, DarA and DarB[4,10,11]. DarA belongs to the large family of PII-like signaling proteins that control a variety of processes, mainly in nitrogen metabolism[12]. The DarB protein consists of a tandem of two CBS (cystathionine-beta synthase) domains, an arrangement called Bateman domain[13]. CBS domains bind AMP, ATP, or other adenosine-derived molecules. CBS domains are present in a variety of proteins, including osmolyte and metal ion transporters, enzymes, and transcription regulators. Recently, CBS domain-containing osmolyte and magnesium transporters were found to bind c-di-AMP. In the case of the osmolyte transporters, the proteins are inactivated upon c-di-AMP binding[4,14,15]. Interestingly, in contrast to most other c-di-AMP-binding proteins, DarA and DarB do not contain any other domain that might be controlled by the binding of the second messenger. It is, therefore, likely that these proteins interact with other proteins in a c-di-AMP-dependent manner to control their activity.

In this study, we performed an unbiased search for potential interaction partners of the DarB protein. This search identified the Rel protein that synthesizes and degrades the alarmone nucleotide (p)ppGpp. The accumulation of this signaling nucleotide results in a global switch off of cellular activities in bacteria, including DNA replication, nucleotide biosynthesis, transcription of household genes, and translation[16,17]. Thus, the integration of c-di-AMP and (p)ppGpp signaling allows a global cellular response to the availability of potassium.

## Results

**Identification of Rel as an interaction partner of DarB**. We assumed that DarB might act by interaction with other proteins. A *L. monocytogenes* strain lacking c-di-AMP is unable to grow on complex media, but suppressor mutants with the inactivated homolog of DarB (CbpB) were able to grow on complex medium[18]. This observation suggests that the apo-form of DarB exerts some harmful interactions. In both *B. subtilis* and *L. monocytogenes*, DarB is encoded in a conserved operon with the transcription factor CcpC, the regulator of the citric acid cycle[19,20]. We hypothesized that DarB might control the activity of CcpC. However, attempts to detect an interaction between the two proteins failed suggesting that DarB exerts a different function (Supplementary Fig. 1).

To get a first unbiased glimpse on the function of DarB, we identified potential interaction partners by passing a *B. subtilis* crude extract over a DarB-saturated column. The proteins were then eluted from the column, and the co-purified proteins were identified by mass spectrometry. In agreement with previous results, CcpC was not identified in the fraction that co-elutes with DarB. In contrast, the analysis identified the GTP pyrophosphokinase Rel as a top scoring protein (Supplementary Data 1). This protein was not detected in the negative control and was therefore considered as a putative interaction partner of DarB. Rel catalyzes the production of the alarmones ppGpp and pppGpp by transferring pyrophosphate derived from ATP to GDP and GTP, respectively, under conditions of amino acid starvation. Moreover, Rel degrades both alarmones if amino acids become available[21].

In order to gain further evidence for the interaction between DarB and Rel, we used the bacterial two-hybrid system in which an adenylate cyclase is reconstituted if cloned proteins interact with each other resulting in β-galactosidase activity. As shown in Fig. 1a, both DarB and Rel exhibited self-interaction, in agreement with structural analysis of these proteins[22]. In addition, co-expression of DarB and Rel resulted in the reconstitution of a functional adenylate cyclase, thus confirming the interaction of the two proteins. None of the two proteins showed an interaction with the Zip protein, which was used as the negative control. Thus, the interaction between DarB and Rel is specific.

Furthermore, we performed size exclusion chromatography-multiangle light scattering (SEC-MALS) experiments with DarB and the purified Rel protein (see Supplementary Fig. 2) to get in vitro confirmation for the interaction. As shown in Supplementary Fig. 3b, the two protein co-elute in vitro. In contrast, no co-elution was detectable when DarB was saturated with c-di-AMP (Supplementary Fig. 3c). This observation suggests that only apo-DarB is capable of interacting with Rel. It is in agreement with the initial pull-down experiment and the bacterial two-hybrid analysis that both revealed an interaction between the two proteins in the absence of c-di-AMP.

To obtain additional evidence for the specificity of the interaction, we mutated the DarB protein in a way to prevent the interaction with Rel. An inspection of the DarB structure (PDB code 1YAV) as well as of the structure of the DarB-c-di-AMP complex (Heidemann and Ficner, unpublished results) suggested that surface residues close to the c-di-AMP binding site might interfere with Rel binding. We exchanged Ala-25 and Arg-132 to Gly and Met, respectively, in single mutants, and combined the two mutations. The resulting DarB$^{A25G,R132M}$ was tested for c-di-AMP and Rel binding. Isothermal titration calorimetry (ITC) experiments indicated that the mutated protein binds c-di-AMP (Supplementary Fig. 4a) demonstrating that the protein folds correctly. However, a SEC-MALS analysis showed that the mutant protein binds much weaker to Rel as compared to the wild type protein. Moreover, this residual interaction is not affected by c-di-AMP (Supplementary Fig. 3d, e).

Taken together, these data indicate that DarB specifically binds to Rel, and that this interaction is inhibited by c-di-AMP.

**Biochemical and physiological regulation of the interaction**. To further investigate the role of c-di-AMP in the interaction between DarB and Rel, we assayed the binding of purified DarB to immobilized Strep-tagged Rel in the absence or presence of c-di-

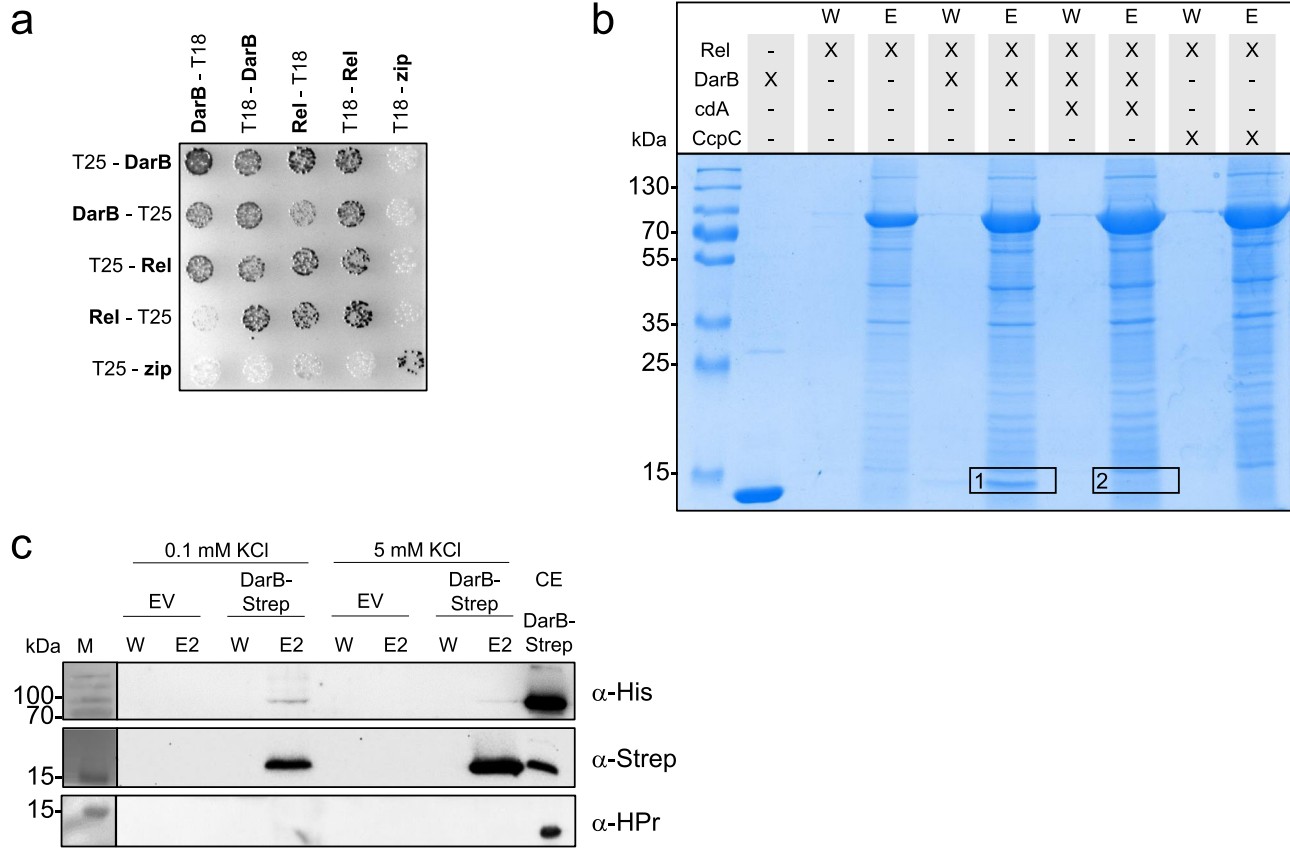

**Fig. 1 DarB interacts with Rel in vitro and in vivo. a** Bacterial two-hybrid (BACTH) experiment testing for the interaction of DarB with Rel. N- and C-terminal fusions of DarB and Rel to the T18 or T25 domains of the adenylate cyclase (CyaA) were created and the proteins were tested for interaction in *E. coli* BTH101. Dark colonies indicate an interaction that results in adenylate cyclase activity and subsequent expression of the reporter β-galactosidase. The experiment was conducted three times and a representative plate is shown. **b** In vitro Strep-Rel pulldown experiment. Strep-Rel was immobilized onto a StrepTactin column and incubated with DarB, DarB preincubated with c-di-AMP, or the control protein CcpC. The eluates (E) and wash (W) fractions were analyzed by SDS-PAGE and the presence of DarB in the elution fractions was further verified by MS analysis (excised gel bands are numbered with 1 and 2). The experiment was conducted three times and a representative gel is shown. **c** In vivo interaction experiment of DarB-Strep with Rel-His. *B. subtilis* expressing Rel-His$_6$ was transformed with plasmid-borne DarB-Strep and grown in minimal medium containing low (0.1 mM) or high (5 mM) potassium concentration. DarB together with its potential binding partners was purified with a StrepTactin column and the elution and wash fractions were analyzed by western blot analysis. DarB and Rel were detected by using antibodies against the Strep-tag and the His-tag, respectively. HPr served as a negative control. EV, empty vector; CE, cell extract; cdA, c-di-AMP. The western blot was conducted three times and a representative gel is shown.

AMP. While DarB was co-eluted with Rel in the absence of c-di-AMP, no DarB was retained on the column when c-di-AMP was present (see Fig. 1b). LC-MS analysis of the gel segments corresponding to the size of DarB confirmed this result (Fig. 1b, segments 1 and 2). No interaction between Rel and the negative control CcpC was detected (Fig. 1b, see also Supplementary Fig. 5, Supplementary Table 1). Similarly, the interaction was abolished if the DarB protein had mutations affecting A25 alone or in combination with the R132M substitution (Supplementary Fig. 6). These results support the specific interaction between Rel and DarB, and they confirm that Rel interacts with the apo-form of DarB but not with the DarB/ c-di-AMP complex.

c-di-AMP is a second messenger that functions in potassium homeostasis, and the intracellular levels of the nucleotide correlate with the potassium concentration[1]. We tested, therefore, how the external potassium supply would affect the interaction between DarB and Rel in vivo. For this purpose, we used a strain that expressed His-tagged Rel from the chromosome and Strep-tagged DarB from a plasmid. This strain was cultivated in minimal medium at low (0.1 mM) and high (5 mM) potassium concentrations, and the protein extract was passed over a StrepTactin column to isolate Strep-DarB in complex with its

potential interaction partners (Fig. 1c). The presence of the Rel protein in the elution fractions was analyzed by a Western blot using antibodies specific for the His-tag. While His-Rel was co-eluted with DarB at the low potassium concentration, no interaction was detected when the bacteria had been cultivated at the high potassium concentration. Again, the presence and absence of Rel in eluates from cultures grown at 0.1 or 5 mM potassium, respectively, was verified by mass spectrometry (Supplementary Data 2). No Rel was detectable in the eluate of the culture grown at the high potassium concentration. Since the intracellular c-di-AMP concentration is low at an external potassium concentration of 0.1 mM[6], we conclude that the interaction between DarB and Rel occurs at low potassium concentrations when c-di-AMP is not bound to DarB. This conclusion is in excellent agreement with the observed inhibition of the interaction by c-di-AMP (see Fig. 1b).

**DarB does not interact with other small alarmone synthetases.** In addition to Rel, *B. subtilis* encodes two additional (p)ppGpp synthesizing enzymes, the small alarmone synthetases SasA and SasB[23]. In contrast to Rel, which is a multidomain protein (see

below), the latter proteins consist of a stand-alone synthetase domain. To test whether these proteins are also capable of interacting with DarB, we made use of the two-hybrid system as described above for Rel. In agreement with the known formation of homotetramers[24], we observed self-interactions for both SasA and SasB. This also indicates that the fusion proteins have folded correctly. Again, we confirmed the interaction between Rel and DarB. However, no interaction of SasA and SasB with DarB could be detected (see Supplementary Fig. 7). The absence of an interaction between DarB and the small alarmone synthetases is supported by the fact that the proteins did not co-elute with Strep-DarB in the in vivo experiments described above. Thus, the interaction of DarB is most likely specific for Rel.

**DarB interacts with the N-terminal portion of Rel.** The Rel protein is a multidomain protein that consists of a N-terminal hydrolase (HYD) domain, the synthetase (SYN) domain, the TGS domain (for: ThrRS, GTPase, and SpoT), a zinc finger domain (ZFD), and the C-terminal RNA recognition motif (RRM) domain (see Fig. 2a)[25]. While the HYD and SYN domains are required for the degradation and synthesis of (p)ppGpp, respectively, the C-terminal domains are involved in the interaction with the ribosome and the control of the enzymatically active domains[25]. To test the contribution of the N- and C-terminal regions of Rel to the interaction with DarB, we analyzed the protein–protein interactions using the bacterial two-hybrid system (see Fig. 2b). The Rel fragment consisting of the SYN domain and the C-terminal regulatory domains showed a very faint interaction with DarB. In contrast, a very strong interaction was observed for the N-terminal fragment consisting of the HYD and the SYN domains (Rel$^{NTD}$). Thus, in contrast to the interaction of Rel with the ribosome which is mediated by the C-terminal RRM domain, DarB seems to bind to the N-terminal part of Rel.

To confirm the binding of the N-terminal region of Rel to DarB, we assayed binding of DarB to the immobilized truncated Rel$^{NTD}$ protein that lacked the C-terminal part. As observed for the full-length protein, this HYD-SYN fragment of Rel bound to DarB, and this interaction was prevented by the addition of c-di-AMP (Fig. 2c, Supplementary Table 1).

In order to confirm the complex formation of Rel$^{NTD}$ and DarB in vitro, a SEC-MALS experiment was performed. The separated elution profiles of the two proteins correspond to a monomer for Rel$^{NTD}$ and a dimer for DarB. A dimer formation by DarB is in agreement with the results from the two-hybrid analysis (Fig. 1a) and the available crystal structure of the apo-protein (PDB 1YAV). By contrast, Rel$^{NTD}$ was unable to exhibit self-interactions in the two-hybrid screen (see Fig. 2b). Co-elution of Rel$^{NTD}$ and DarB resulted in an earlier eluting peak, indicating the formation of a complex of 94.2 kDa consisting of DarB and Rel$^{NTD}$ (see Fig. 2d, Supplementary Fig. 8). The subsequent SDS-page analysis of the elution fractions confirmed that both proteins co-eluted from the column (see Supplementary Fig. 8). To determine the kinetic parameters of the interaction, we performed isothermal titration calorimetry (ITC) experiments (Fig. 2e). Titration of DarB against Rel$^{NTD}$ revealed an equimolar stoichiometry of the two proteins in the complex. Moreover, we determined the affinities of DarB for c-di-AMP and Rel. While the $K_D$ for the binding of c-di-AMP was about 45 nM (Supplementary Fig. 4a, b), we observed a $K_D$ of 650 nM for the interaction of DarB and Rel (see Fig. 2e). This about 15-fold higher affinity of DarB for c-di-AMP is crucial for the c-di-AMP-mediated regulation of the DarB–Rel interaction.

**Genetic support for the DarB–Rel interaction.** So far, no function other than binding to c-di-AMP and to Rel has been

identified for DarB. To get better insights into the physiological role(s) of DarB, we constructed strains that either lacked DarB (ΔdarB, GP3409) or that overexpressed the protein (darB$^+$, 168 + pGP3306) and compared growth of the three strains in minimal medium with 0.1 or 5 mM potassium (see Fig. 3a). All three strains grew very similar with 5 mM potassium (growth rates of 0.43, 0.39, and 0.43 h$^{-1}$ for the wild type, the darB$^+$, and the ΔdarB mutant, respectively). In contrast, at the low potassium concentration, we observed a delayed growth for the strain overexpressing DarB as compared to the wild type and the darB deletion mutant (0.09 vs. 0.21 and 0.18 h$^{-1}$, respectively). This is the condition when c-di-AMP is present in low amounts, and thus a large fraction of the DarB protein is present as apo-protein with the capacity to bind to Rel. It is, therefore, tempting to speculate that this interaction might be the reason for the growth defect. To test this idea, we deleted the rel gene in the wild type strain and in the strain carrying the expression vector for darB and compared the growth in minimal medium at a low potassium concentration. As shown in Fig. 3b, the deletion of rel suppressed the growth defect that resulted from the overexpression of DarB (growth rates of 0.15 and 0.18 h$^{-1}$ for the rel mutant and the rel mutant with overexpression of DarB, respectively). In contrast, deletion of the rel gene did not affect the growth of B. subtilis if compared to an isogenic wild type strain on minimal medium containing ammonium and 0.1 mM potassium (Supplementary Fig. 9). Overexpression of the DarB mutant proteins that are defective in the interaction with Rel did not result in growth inhibition (Fig. 3c). Taken together, all these observations indicate that the growth-inhibiting effect of DarB overexpression is the result of its interaction with Rel and suggests that DarB might control Rel activity.

Transcription of rRNA promoters is decreased under conditions of the stringent response[26,27]. We observed that RNA extracted from the strain overexpressing DarB, in contrast to the wild type, lacked the rRNA intermediate migrating above the 16S band (Supplementary Fig. 10). This band corresponds to the size of the pre-16S rRNA[28]. This decrease in the pre-16S rRNA pool most likely results from the reduced rRNA promoter activity and drainage of the pre-RNA pool. This suggests that the overexpression of DarB affects rrn transcription by stimulation of (p)ppGpp synthesis in vivo. The downregulation of rRNA genes results in problems in ribosome assembly and might explain the observed Rel-dependent growth defect of the DarB overexpression strain during exponential growth.

**DarB controls Rel synthetase and hydrolase activities.** The results presented above suggest that the interaction between DarB and Rel might affect synthesis of (p)ppGpp by Rel. To test this idea, we used purified Rel protein to assay its synthetase and hydrolase activities. The purified Rel protein had little biosynthetic activity, as indicated by the production of 1.1 pmol pppGpp per pmol of Rel per minute (Fig. 4a). This is in good agreement with the absence of Rel synthetase activity if not triggered by uncharged tRNAs at the ribosome[25]. In contrast, Rel activity was enhanced threefold if purified DarB protein was added to the assay mixture. If DarB was saturated with c-di-AMP prior to incubation with Rel, the Rel protein retained its background activity and no enhancement was detected. The addition of c-di-AMP to Rel did not affect pppGpp synthesis indicating that the reduced Rel biosynthetic activity in the presence of the DarB–c-di-AMP complex as compared to apo-DarB is not the result of Rel inhibition by c-di-AMP but indeed reflects the loss of Rel activation by DarB in the presence of c-di-AMP. No pppGpp synthesis was detected with DarB alone indicating that DarB is unable to synthesize pppGpp and that the addition of DarB

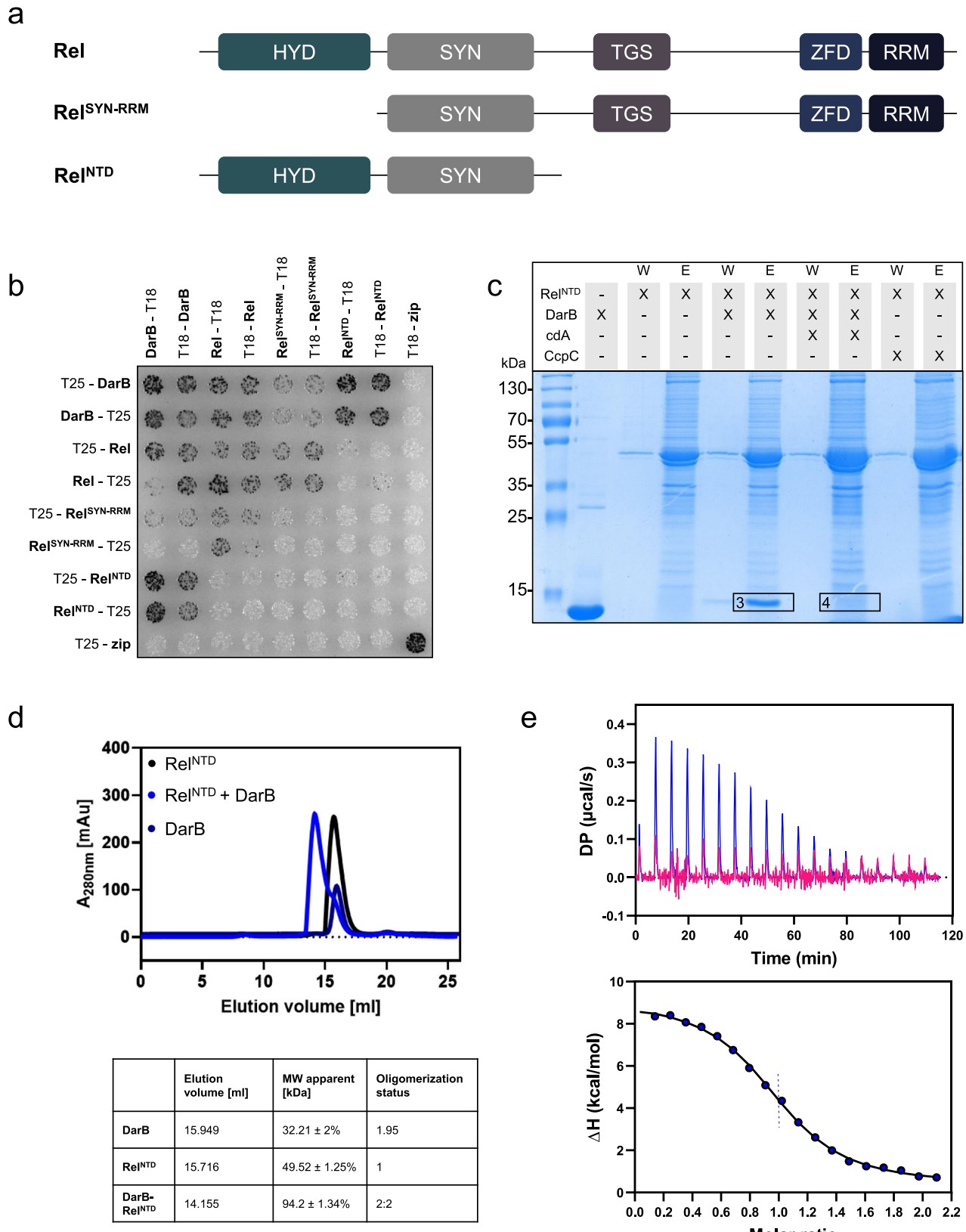

activates the synthetase activity of Rel. To exclude the possibility that the activation was just a result of non-specific protein crowding, we also assayed pppGpp synthesis by Rel in the presence of bovine serum albumin. In this case, Rel exhibited only background activity (see Fig. 4a). These results demonstrate that the interaction between apo-DarB and Rel stimulates the synthesis of pppGpp and that this stimulation is prevented in the presence of c-di-AMP.

**Fig. 2 DarB binds the N-terminal domain (NTD) of Rel. a** The domain organization of Rel and the truncated Rel variants used in this study. HYD, hydrolase domain; SYN, synthetase domain; TGS, TGS domain (for: ThrRS, GTPase, and SpoT); ZFD, a zinc finger domain; RRM domain (for RNA Recognition Motif). **b** Bacterial two-hybrid (BACTH) assay to test for the interaction between DarB and the full-length and truncated Rel-variants. N- and C-terminal fusions of DarB and the Rel variants to the T18 or T25 domain of the adenylate cyclase (CyaA) were created and the proteins were tested for interaction in *E. coli* BTH101. Dark colonies indicate an interaction that results in adenylate cyclase activity and subsequent expression of the reporter β-galactosidase. The experiment was conducted three times and a representative plate is shown. **c** In vitro pulldown experiment with the NTD of Rel. Strep-Rel$^{NTD}$ was immobilized onto a StrepTactin column and incubated with DarB, DarB preincubated with c-di-AMP, or the control protein CcpC. The eluate and wash fractions were analyzed by SDS-PAGE and the presence of DarB in the elution fractions was further verified by MS analysis (excised gel bands are numbered with 3 and 4). The experiment was conducted three times and a representative gel is shown. **d** The DarB–Rel$^{NTD}$ complex was analyzed by size exclusion chromatography and multi-angle light scattering (SEC-MALS). Rel$^{NTD}$ and DarB were used in equimolar concentrations. Dark blue line, DarB; black line, Rel$^{NTD}$; blue line, mixture of DarB and Rel. The calculated molar masses determined by MALS are listed below the chromatogram. **e** The molar ratio of the DarB–Rel$^{NTD}$-complex was assessed by Isothermal titration calorimetry (ITC). The cell and the syringe contained 10 µM Rel$^{NTD}$ and 100 µM DarB (blue) or 100 µM c-di-AMP-bound DarB (DarB$^{cdA}$) (pink), respectively. cdA, c-di-AMP. Source data are provided as a Source data file.

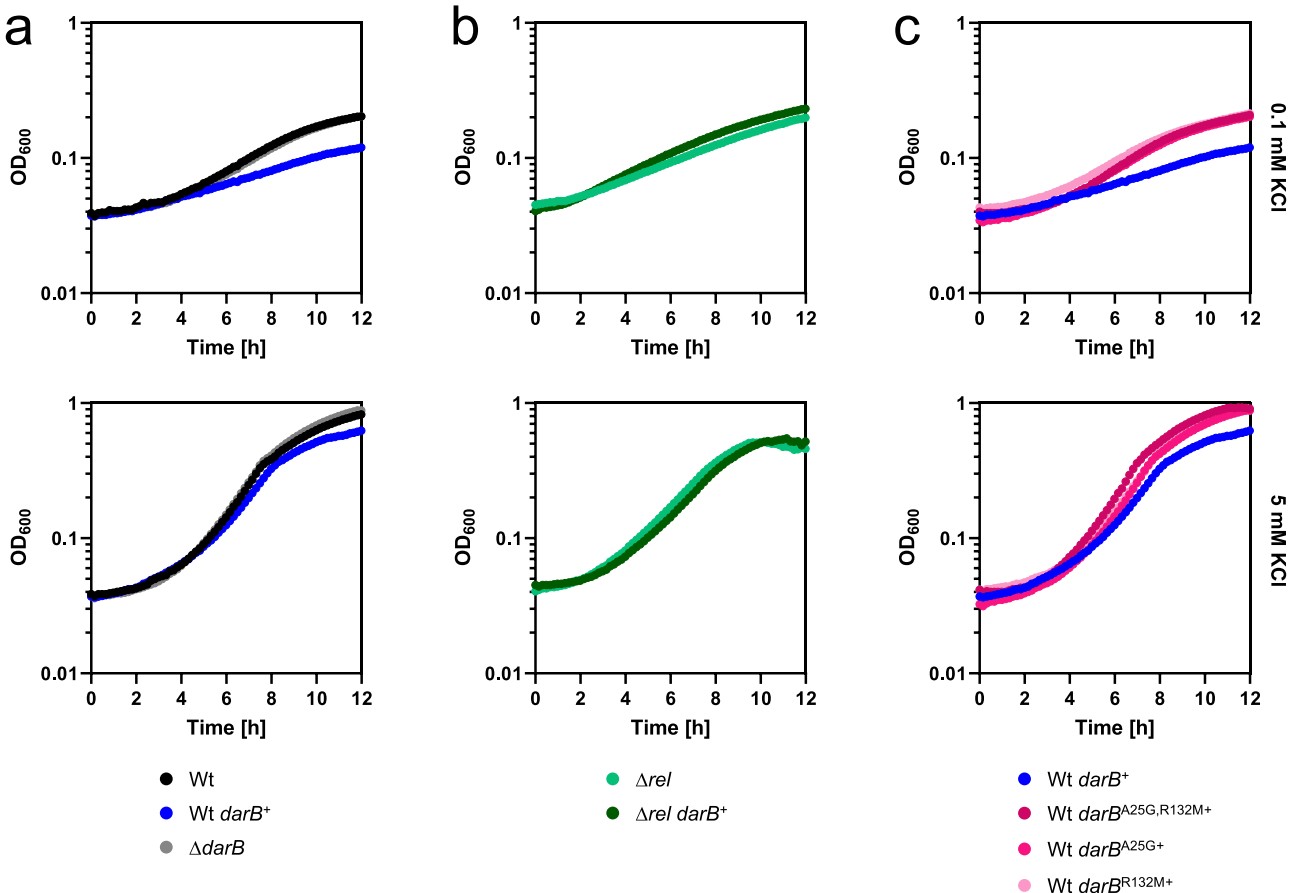

**Fig. 3 Overexpression of DarB is toxic.** Growth experiments of **a** wild type *B. subtilis* (black), GP3407 (Δ*darB*; gray), wild type + pGP3306 (*darB$^+$*, blue). **b** GP3419 (Δ*rel*, bright green) and GP3419 (Δ*rel, darB$^+$*, dark green), and **c** wild type + pGP3437/3441/3601 (*darB$^{A25G}$, darB$^{R132M}$, darB$^{A25G,R132M}$*, pink) in MSSM minimal medium with 0.1 mM KCl (upper panel) or 5 mM KCl (lower panel). Growth was monitored in an Epoch 2 Microplate Spectrophotometer (BioTek Instruments) at 37 °C with linear shaking at 237 cpm (4 mm) for 12 h. The growth experiment was conducted three times and a representative curve is shown. Source data are provided as a Source data file.

We also tested whether DarB affects the hydrolase activity of Rel. In this case, we determined the formation of GTP resulting from the hydrolysis of pppGpp. For the Rel protein alone, we determined a turnover rate of 8.7 pmol of GTP per pmol of Rel per minute (Fig. 4b). In the presence of DarB, this activity was reduced six-fold. In the presence of c-di-AMP, DarB has little effect on the pppGpp hydrolase activity of Rel. As observed for the synthetase activity, these effects are specific since DarB has no pppGpp hydrolase activity, c-di-AMP alone does not affect the hydrolase activity of Rel, and the control protein (BSA) does not

affect the hydrolytic activity of Rel. Taken together, these data demonstrate that DarB affects Rel activity by stimulation and inhibition of (p)ppGpp synthesis and degradation, respectively.

The data presented above suggest that the interaction of DarB with Rel results in a net increase of the intracellular (p)ppGpp levels. To verify this assumption, we compared the intracellular (p)ppGpp concentrations in a wild type strain, the *darB$^+$* strain overexpressing DarB, and in the *darB* mutant (Fig. 4c). Indeed, overexpression of DarB resulted in a significant increase of (p)ppGpp (95 pmol OD$_{600}$$^{-1}$ ml$^{-1}$ vs. 59 pmol OD$_{600}$$^{-1}$ ml$^{-1}$). In

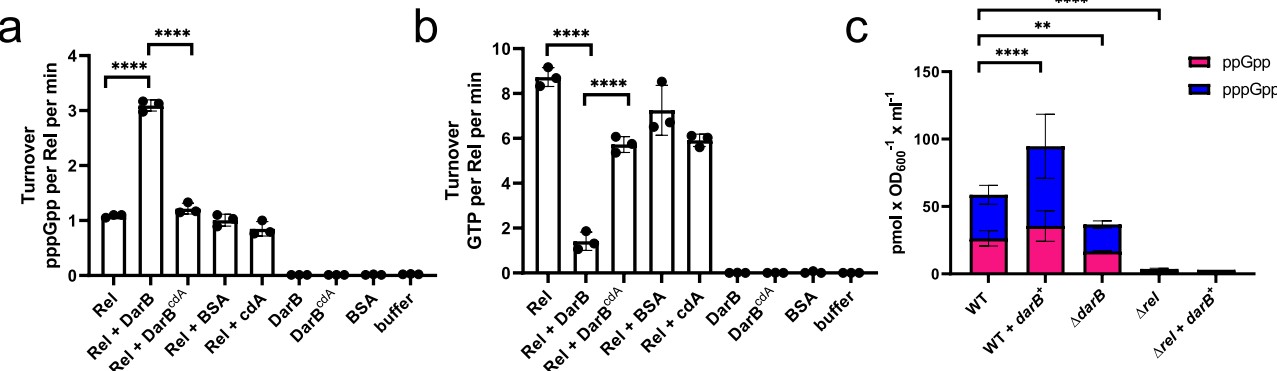

**Fig. 4 DarB stimulates Rel-dependent accumulation of pppGpp.** The activity of Rel was assessed in an in vitro activity assay. **a** Rel synthetase activity assay. Purified Rel (0.5 µM) was incubated with ATP and GTP, in the absence or presence of DarB, c-di-AMP-saturated DarB (5 µM), or bovine serum albumin (BSA) (5 µM) and the production of pppGpp was determined by liquid chromatography coupled tandem mass spectrometry on a QTRAP 5500 instrument (Sciex, Framingham, Massachusetts) equipped with an electrospray ionization source (ESI). BSA served as a negative control. **b** Rel hydrolase activity assay. Purified Rel (0.5 µM) was incubated with pppGpp, in the absence or presence of DarB, c-di-AMP-saturated DarB (5 µM), or bovine serum albumin (BSA) (5 µM) and the production of GTP was monitored. The enzymatic assays were conducted with $n = 3$ biologically independent samples. Data are presented as mean values ± SD. Statistical analysis was performed using a one-way ANOVA, followed by Tukey's multiple comparisons test (**** $P < 0.0001$). **c** Determination of intracellular (p)ppGpp levels in wild type *B. subtilis*, GP3407 (Δ*darB*), wild type + pGP3306 (*darB*+), GP3419 (Δ*rel*) and GP3419 + pGP3306 (Δ*rel, darB*+). Bacteria were grown in MSSM minimal medium with 0.1 mM KCl until the exponential growth phase, and the nucleotides were analyzed as described above. The experiment was conducted with $n = 3$ biologically independent samples (in the case of wild type + pGP3306 (*darB*+), 6 biologically independent were analyzed). Data are presented as mean values ± SD. Statistical analysis was performed using a one-way ANOVA, followed by Tukey's multiple comparisons test (**** $P < 0.0001$). Source data are provided as a Source data file. cdA, c-di-AMP; DarB$^{cdA}$, DarB saturated with c-di-AMP.

the *darB* mutant, the (p)ppGpp concentration was reduced as compared to the wild type strain. The determination of the (p)ppGpp concentration in a *rel* mutant strain revealed that Rel was the major source of (p)ppGpp production under the conditions of our experiment, and that the residual (p)ppGpp synthesis was not affected by DarB overexpression. These observations are in agreement with the physiological observations. They confirm that the accumulation of (p)ppGpp is the cause of the growth inhibition of the strain that overexpresses DarB.

## Discussion

In this work, we report a link between potassium concentration, c-di-AMP signaling, and the stringent response in the Gram-positive model organism *B. subtilis*. The observation that at low external potassium and intracellular c-di-AMP concentrations, the apo form of DarB binds to the alarmone synthetase Rel and triggers (p)ppGpp accumulation independent of the ribosome complements the earlier observations that on the other hand (p)ppGpp binds to the c-di-AMP degrading phosphodiesterases GdpP and PgpH to inhibit the degradation of c-di-AMP[29–31]. Together, these data allow to develop a model (Fig. 5a) in which at low potassium concentrations the intracellular c-di-AMP levels are low and the c-di-AMP targets including DarB are present in the apo form. DarB then binds to Rel and stimulates the synthesis of the alarmone (p)ppGpp in a ribosome-independent manner. The accumulation of (p)ppGpp results in a re-organisation of cellular physiology including the stop of translation. This direct link between the potassium concentration, the stringent response, and ribosome activity is very important for the cell since potassium is essential for ribosome assembly and translation at the ribosome[32,33]. On the other hand, the accumulation of (p)ppGpp interferes with the degradation of c-di-AMP. This is likely to be important if potassium becomes available again. Then, c-di-AMP synthesis can be initiated and as long as (p)ppGpp is present, the second messenger is protected from degradation. This allows to achieve a c-di-AMP concentration that is appropriate to adjust

the cellular potassium homeostasis by binding to c-di-AMP responsive riboswitches that control the expression of high affinity potassium transporters as well as to the potassium importers and exporters to inhibit and activate these proteins, respectively.

There is a huge body of evidence that (p)ppGpp synthesis by Rel is triggered by uncharged tRNA in the ribosomal A-site upon amino acid starvation[17,25]. Our work supports the idea of ribosome-independent stimulation of the stringent response, as shown for phosphate and fatty acid starvation in *E. coli*[34]. Our work now extends this concept also to potassium starvation in *B. subtilis*. Similar to the presence of uncharged tRNAs, a lack of potassium results in a stop of translation[33], and does thus require similar global responses to reprogram translation, gene expression, DNA replication, and cellular metabolism. It is interesting to note that c-di-AMP is a second messenger that reports on potassium availability in most Gram-positive and also in many Gram-negative bacteria with the notable exception of α-, β-, and γ-proteobacteria[7,8]. In α- and β-proteobacteria, a regulatory protein, PtsN (also referred to as enzyme IIA$^{Ntr}$) is capable of interacting with the single (p)ppGpp synthetase/hydrolase of these bacteria. The interaction depends on the nitrogen supply and the resulting phosphorylation state of PtsN and leads to the accumulation of (p)ppGpp[35–37]. PtsN has also been implicated in the control of potassium homeostasis: in *E. coli*, non-phosphorylated PtsN binds and inhibits TrkA, a subunit of low-affinity potassium transporters as well as to the two-component sensor kinase KdpD, thus stimulating its activity and the expression of the high-affinity Kdp potassium transport system[38,39]. It is thus intriguing to speculate that the regulatory link between potassium homeostasis and the stringent response is conserved in bacteria even though the specific molecular mechanisms may be completely different.

An interesting aspect of this study is the mode of DarB regulation by c-di-AMP. Our biophysical interaction analyses indicated an equimolar stoichiometry of the two proteins. Since DarB forms dimers, and the mutations that interfere with Rel binding

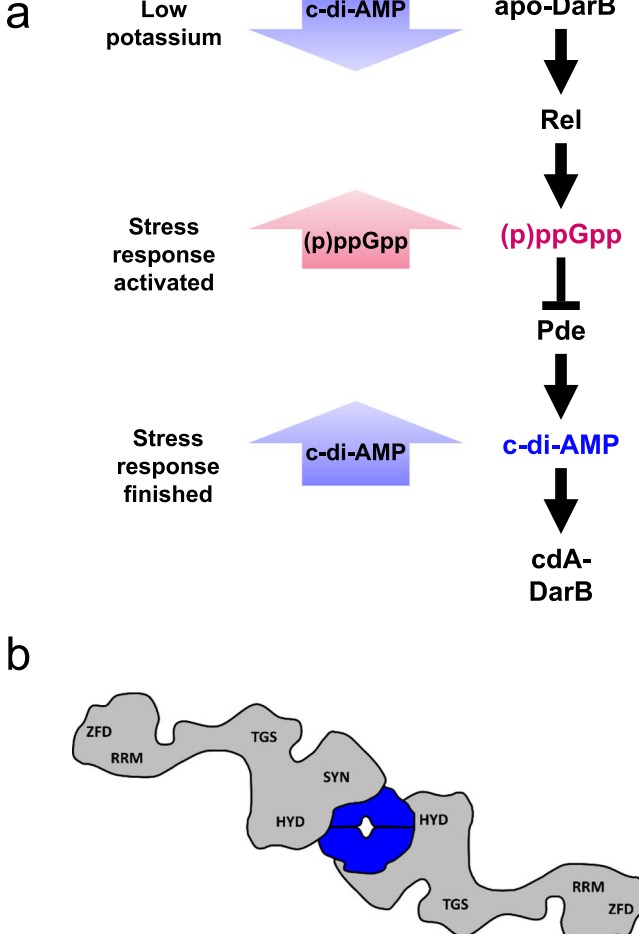

**a**

Low potassium → c-di-AMP → apo-DarB

Rel

Stress response activated → (p)ppGpp → (p)ppGpp

Pde

Stress response finished → c-di-AMP → c-di-AMP

cdA-DarB

**b**

**Fig. 5 The link between c-di-AMP and (p)ppGpp signaling in _B. subtilis_.** **a** The model depicts the bidirectional and dynamic process of the cellular response to potassium limitation. When potassium becomes limiting, the diadenylate cyclases respond and produce less c-di-AMP and c-di-AMP receptor proteins are present in the apo-form. One apo-DarB dimer binds to Rel and by this disrupts the Rel dimer. This leads to formation of the DarB–Rel complex as suggested by the presented data. One DarB dimer is bound by two Rel monomers, one on each side. The interaction occurs via the HYD-SYN domains of Rel. Interaction of DarB with Rel leads to stimulation of (p)ppGpp synthesis. (p)ppGpp accumulation induces the stringent response and inhibits the c-di-AMP-degrading phosphodiesterases GdpP and PgpH. This leads to increasing intracellular c-di-AMP amounts. DarB can then bind c-di-AMP and is thus no longer able to interact with Rel. **b** The model shows the DarB-Rel complex as suggested by our data. One DarB dimer is bound by two Rel monomers, one on each side. The interaction occurs via the HYD-SYN domains of Rel. DarB, blue; Rel, gray. cdA, c-di-AMP; HYD, hydrolase domain; SYN, synthetase domain; TGS, TGS domain (for: ThrRS, GTPase, and SpoT); ZFD, Zinc finger domain; RRM, RNA Recognition Motif.

are located at the upper and lower side of the DarB dimer, it is tempting to speculate that the proteins form a sandwich-like complex with a central DarB dimer and a molecule of Rel on each face of the dimer (see Fig. 5b). This resulting 2:2 stoichiometry is also best compatible with the results of the SEC-MALS analysis. The physiological relevance of the interaction is not only supported by the Rel-dependent growth inhibition upon overexpression of DarB, but also by the intracellular concentrations of both proteins (0.6 and 0.9 μM for Rel and DarB, respectively)[40]

which are in the range of the $K_D$ determined for their interaction. Moreover, the differential affinities of DarB to c-di-AMP and Rel as well as the fact that two molecules of c-di-AMP bind to each DarB dimer in the region that is also important for Rel binding (Heidemann and Ficner, unpublished results) suggest that c-di-AMP and Rel compete for DarB binding. Since c-di-AMP has a 15-fold higher affinity for DarB than Rel, it is tempting to speculate c-di-AMP inhibits Rel binding in a competitive manner.

For all other processes controlled by c-di-AMP as well as other second messengers such as c-di-GMP, the nucleotide directly binds to its targets to control their activity, as has been shown for potassium uptake or export, osmolyte export, or pyruvate carboxylase activity[4,11,14,15,41]. This raises the question why Rel needs DarB as a mediator of c-di-AMP mediated control. First, Rel is already composed of multiple domains, and it might have been difficult in evolution to integrate a further level of signaling directly into the protein. Second, potassium starvation is completely different from, but as serious for the cell as amino acid starvation. This makes it advantageous to have the two regulatory pathways for Rel activity separated from each other. Moreover, it is the apo form of DarB that binds and regulates Rel activity. An important function for apo-DarB has already been suggested by the observation that a _L. monocytogenes_ mutant lacking c-di-AMP readily acquires mutations affecting the DarB counterpart CbpB[18]. Similarly, mutations inactivating the DarA ortholog PstA were found in _L. monocytogenes_[18] suggesting that this protein might also interact with its partners in the apo form under conditions of potassium starvation.

DarB is conserved in several Gram-positive bacteria, including _L. monocytogenes_ and _Enterococcus faecalis_. Indeed, while this study has been under consideration, control of Rel activity by the _L. monocytogenes_ DarB counterpart (CbpB) has been reported[42]. In addition to DarB, _B. subtilis_ Rel has been shown to interact with the competence protein ComGA, resulting in the inhibition of the hydrolase activity of Rel[43]. Moreover, a recent study demonstrated the transient accumulation of (p)ppGpp upon heat stress in _B. subtilis_[44]. It will be interesting to study whether yet additional factors may control Rel activity to trigger the stringent response under specific stress conditions and whether and how interaction with the ribosome affects the outcome of the Rel–DarB interaction.

## Methods

**Strains, media, and growth conditions.** _E. coli_ DH5α and Rosetta DE3[45] were used for cloning and for the expression of recombinant proteins, respectively. All _B. subtilis_ strains used in this study are derivatives of the laboratory strain 168. _B. subtilis_ and _E. coli_ were grown in Luria-Bertani (LB) or in sporulation (SP) medium[45,46]. For growth assays and the in vivo interaction experiments, _B. subtilis_ was cultivated in MSSM medium[6]. In this medium $KH_2PO_4$ was replaced by $NaH_2PO_4$ and KCl was added as indicated. The media were supplemented with ampicillin (100 μg/ml), kanamycin (50 μg/ml), chloramphenicol (5 μg/ml), or erythromycin and lincomycin (2 and 25 μg/ ml, respectively) if required.

**Phenotypic characterization.** To assay growth of _B. subtilis_ mutants at different potassium concentrations, the bacteria were inoculated in LB medium and precultured in MSSM medium with 0.1 mM KCl. The cultures were grown until exponential phase, harvested, washed three times in MSSM basal salts solution before an optical density at 600 nm ($OD_{600}$) was adjusted to 1.0. For growth analysis in liquid medium, the cells were used to inoculate a 96 well plate (Microtest Plate 96 Well, Sarstedt) containing MSSM medium with ammonium and the required potassium concentrations. Growth was tracked in an Epoch 2 Microplate Spectrophotometer (BioTek Instruments) at 37 °C with linear shaking at 237 cpm (4 mm) for 20 h, and an $OD_{600}$ was measured in 10 min intervals. For growth analysis on solid medium, cells were grown, harvested, and washed as described above. The cell density was adjusted to $OD_{600}$ of 1.0 in MSSM basal salts solution. Dilution series were prepared and then spotted onto MSSM plates with ammonium or glutamate and different KCl concentrations, or complex medium.

**DNA manipulation.** Transformation of _E. coli_ and plasmid DNA extraction were performed using standard procedures[45]. All commercially available plasmids,

restriction enzymes, T4 DNA ligase and DNA polymerases were used as recommended by the manufacturers. Chromosomal DNA of *B. subtilis* was isolated using the Bacterial DNA kit (PeqLab, Erlangen, Germany)[46]. *B. subtilis* was transformed with plasmid and genomic DNA according to the two-step protocol[46]. Introduction of mutations in the *darB* allele was achieved by the Combined Chain Reaction by using an additional 5′ phosphorylated primer to introduce the mutation[47].

**Construction of mutant strains by allelic replacement.** Deletion of the *darB* and *rel* genes was achieved by transformation of *B. subtilis* 168 with a PCR product constructed using oligonucleotides to amplify DNA fragments flanking the target genes and an appropriate intervening resistance cassette[48] (Supplementary Table 3). The integrity of the regions flanking the integrated resistance cassette was verified by sequencing PCR products of about 1100 bp amplified from chromosomal DNA of the resulting mutant strains, GP3409 and GP3419, respectively. To verify the identity of the *rel* mutant, growth properties of the constructed strain GP3419 were compared to the *rel* mutants BHS126 and BKK27600[49,50] (see Supplementary Fig. 11). A strain allowing expression of Rel fused to C-terminal His-tag was constructed by first generating an appropriate PCR product and subsequent transformation of *B. subtilis* 168. The resulting strain was GP3429.

**Plasmid constructions.** The *ccpC*, *darB*, *rel*, *sasA*, and *sasB* alleles were amplified using chromosomal DNA of *B. subtilis* 168 as the template and appropriate oligonucleotides that attached specific restriction sites to the fragment. Those were: KpnI and BamHI for cloning *rel* in pGP172[51], BamHI and SalI for cloning *rel* in pWH844[52], XbaI and KpnI for cloning all genes in the BACTH vectors[53], BamHI and KpnI sites for cloning *rel* into pGP888[54] for genomic integration. The truncated *rel* variants were constructed as follows: *rel*-SYN-RRM contained aa 168–734, *rel*-HYD-SYN contained aa 1–391. For the overexpression of DarB, *darB* was amplified using chromosomal DNA of *B. subtilis* 168 as the template and appropriate nucleotides that attached BsaI and XhoI restriction sites to the fragments and cloned between the BsaI and XhoI sites of the expression vector pET-SUMO (Invitrogen, Germany). The resulting plasmid was pGP2972. All plasmids and oligonucleotides are listed in Supplementary Tables 2, 3, respectively.

**Protein expression and purification.** *E. coli* Rosetta(DE3) was transformed with the plasmid pGP2972, pGP3437, pGP3441, pGP3460 encoding wild type or mutant 6xHis-SUMO-DarB for purification of DarB or with the plasmids pGP3348 or pGP3350 for expression of Strep-tagged full-length Rel and Rel[NTD], respectively, or pGP706[20] for expression of 6xHis-CcpC. For purification of 6xHis-SUMO-Rel, pVHP186[21] was transformed into *E. coli* Rosetta(DE3). Expression of the recombinant proteins was induced by the addition of isopropyl 1-thio-β-D-galactopyranoside (final concentration, 1 mM) to exponentially growing cultures (OD$_{600}$ of 0.8) of *E. coli* carrying the relevant plasmid. His-tagged proteins were purified in 1× ZAP buffer (50 mM Tris-HCl, 200 mM NaCl, pH 7.5), if not stated otherwise, and Strep-tagged proteins in buffer W (100 mM Tris-HCl, 150 mM NaCl, 1 mM Na$_2$EDTA, pH 8.0). 10×His-SUMO-Rel was purified in buffer A (750 mM KCl, 5 mM MgCl$_2$, 40 μM MnCl$_2$, 40 μM Zn(OAc)$_2$, 20 mM imidazole, 10% glycerol, 4 mM β-mercaptoethanol, 25 mM HEPES:KOH pH 8)[21]. Cells were lysed by four passes at 18,000 p.s.i. through an HTU DIGI-F press (G. Heinemann, Germany). After lysis, the crude extract was centrifuged at 100,000 × g for 60 min and then passed over a Ni$^{2+}$nitrilotriacetic acid column (IBA, Göttingen, Germany) for 6xHis-tagged and 10×His-tagged proteins, or a StrepTactin column (IBA, Göttingen, Germany) for purification of Strep-tagged proteins. The protein was eluted with an imidazole gradient or D-desthiobiotin (2.5 mM), respectively. After elution, the fractions were tested for the desired protein using SDS-PAGE. For the purification of Rel, the column was washed with 8 column volumes of 4 M NaCl to remove RNA prior to elution of the protein with 100 mM and 250 mM imidazole. To remove the SUMO tag from the proteins, the relevant fractions were combined, and the SUMO tag was removed with the SUMO protease (ratio 100:1) during overnight dialysis against 1 x ZAP buffer for DarB or against storage buffer (720 mM KCl, 5 mM MgCl$_2$, 50 mM arginine, 50 mM glutamic acid, 10% glycerol, 4 mM β-mercaptoethanol, 25 mM HEPES:KOH pH 8)[21] for Rel. The cleaved SUMO moiety and the protease were removed using a Ni$^{2+}$nitrilotriacetic acid column (IBA). The purified Rel was concentrated in a Vivaspin turbo 15 (Sartorius) centrifugal filter device (cut-off 50 kDa). The protein was loaded on a HiLoad 16/600 Superdex 200 pg column pre-equilibrated with storage buffer and the fractions containing pure Rel protein were collected and concentrated in a Vivaspin turbo 15 (Sartorius). The purity of protein preparations and the absence of RNA were assessed by SDS-PAGE and on a 1% agarose gel (in 1× TAE buffer; 40 mM Tris-base, 1% acetic acid, 1 mM EDTA pH 8.0), respectively. The protein samples were stored at −80 °C until further use (but no longer than 3 days). The protein concentration was determined according to the method of Bradford[55] using the Bio-Rad dye binding assay and bovine serum albumin as the standard.

**Initial pulldown for identification of potential binding partners.** In order to identify potential binding partners of DarB, *E. coli* Rosetta (DE3) was transformed with pGP2972 (6×His-SUMO-DarB) or the empty vector control pET-SUMO and the protein was overexpressed and purified as described above until the step where the protein was bound to the Ni$^{2+}$nitrilotriacetic acid column. After extensive

washing, *B. subtilis* 168 crude extract (from LB) was added to the column to allow binding of *B. subtilis* proteins to the DarB protein (apo-DarB due to overexpression in *E. coli*). Again, after extensive washing, DarB, together with its potential binding partners, was eluted from the column with an imidazole gradient. The elution fractions from the eluates were subjected to mass spectrometry analysis.

**In vivo detection of protein–protein interactions.** To detect interaction partners of the DarB in vivo, cultures of *B. subtilis* GP3429 containing either pGP767 (DarB-Strep), or the empty vector control (pGP382), were cultivated in 500 ml MSSM medium containing the indicated potassium concentrations until exponential growth phase was reached (OD$_{600}$ ~ 0.4–0.6). The cells were harvested immediately and stored at −20 °C. The Strep-tagged protein and its potential interaction partners were then purified from crude extracts using a StrepTactin column (IBA, Göttingen, Germany) and D-desthiobiotin as the eluent. The eluted proteins were separated on an SDS gel and potential interacting partners were analyzed by staining with Colloidal Coomassie and Western blot analysis. The eluents were further analyzed by mass spectrometry analysis.

**Protein identification by mass spectrometry.** Excised polyacrylamide gel pieces of protein bands were digested with trypsin[56]. Peptides were purified using C18 stop and go extraction (stage) tips[57,58]. Dried peptide samples were reconstituted in 20 μl LC-MS sample buffer (2% acetonitrile, 0.1% formic acid). 2 μl of each sample were subjected to reverse phase liquid chromatography for peptide separation using an RSLCnano Ultimate 3000 system (Thermo Fisher Scientific). Peptides were loaded on an Acclaim® PepMap 100 pre-column (100 μm × 2 cm, C18, 3 μm, 100 Å; Thermo Fisher Scientific) with 0.07% trifluoroacetic acid. Analytical separation of peptides was done on an Acclaim® PepMap RSLC column (75 μm × 50 cm, C18, 3 μm, 100 Å; Thermo Fisher Scientific) running a water-acetonitrile gradient at a flow rate of 300 nl/min. All solvents and acids had Optima grade for LC-MS (Fisher Scientific). Chromatographically eluting peptides were on-line ionized by nano-electrospray (nESI) using the *Nanospray Flex Ion Source* (Thermo Scientific) and continuously transferred into the mass spectrometer (*Q Exactive HF*, Thermo Scientific). Full scans in a mass range of 300–1650 *m/z* were recorded with the *Q Exactive HF* at a resolution of 30,000 followed by data-dependent top 10 HCD fragmentation at a resolution of 15,000 (dynamic exclusion enabled). LC-MS method programming and data acquisition was performed with the XCalibur software 4.0 (Thermo Fisher Scientific).

**LC-MS data analysis.** MS/MS2 data were searched against a *B. subtilis* specific protein database (UniProt Proteome ID UP000001570) using the *Proteome Discoverer Software 2.2*. The digestion mode was trypsin/P, and the maximum of missed cleavage sites was set two. Carbamidomethyl at cysteines was set as fixed modification, and oxidation at methionines and N-terminal acetylation of proteins as variable modifications. Mass tolerances of precursors and fragment ions were 10 ppm and 20 ppm, respectively. False discovery rates were calculated using the reverse-decoy mode, and the filter for peptide spectrum matches was set to 0.01.

**In vitro analysis of protein–protein interactions.** To study the interaction between DarB and Rel, *E. coli* Rosetta (DE3) was transformed with pGP2972 (6×His-SUMO-DarB), pGP3444 (6×His-SUMO-DarB$^{A25G}$), pGP3448 (6×His-SUMO-DarB$^{R132M}$), pGP3460 (6×His-SUMO-DarB$^{A25G,R132M}$), pGP3348 (Strep-Rel), or pGP3350 (Strep-Rel$^{NTD}$), respectively, and the proteins were overexpressed as described above. For purification of Rel or Rel$^{NTD}$, the crude extract was passed over a StrepTactin column (IBA, Göttingen, Germany) and washed with buffer W (pH 8.5) until the wash fractions appeared clear (confirmation with Bradford assay). Purified DarB, if stated preincubated 30 min with c-di-AMP (4× excess), or the control protein CcpC were added to the column, incubation happened overnight at 4 °C under constant rotation. Purification was continued by extensive washing of the column with buffer W before Rel, together with binding partners, was eluted with D-desthiobiotin. For verification of the presence of DarB in the elution fractions, fixed and stained gel bands were excised and submitted to mass spectrometry analysis.

**SEC-MALS.** The interaction of the full-length Rel protein or Rel$^{NTD}$ with DarB or DarB$^{A25G,R132M}$ was analyzed by size-exclusion chromatography and multi-angle light scattering (SEC-MALS). For this purpose, the purified tag-free proteins were either alone or pre-mixed in a 1:1 ratio in storage buffer (for full-lenght Rel) or 1× ZAP buffer (for Rel$^{NTD}$) (~1 mg/ml each) applied onto the column. The buffer was filtered (0.1 μm filters) and degassed in line (Model 2003, Biotech AB/Sweden) prior to protein separation on a S200 Superdex 10/300GL column on an Äkta Purifier (both GE Healthcare). Subsequently, the eluate was analyzed in line with a miniDawn Treos multi angle light scattering system followed by an Optilab T-rEX RI detector (both from Wyatt Technology, Europe) before fractionation. The elution fractions were analyzed with SDS-PAGE. Data analysis was performed using the ASTRA 6.1 software (Wyatt Technology) and also compared to a gel filtration standard (Bio-Rad).

**Determination of binding affinities and of the stoichiometry of the DarB–Rel$^{NTD}$ complex by isothermal titration calorimetry**. ITC experiments were carried out with a VP-ITC microcalorimeter (MicroCal Inc., Northampton, MA) in order to determine the affinity of DarB to Rel$^{NTD}$ and the oligomerization state of the complex. In a typical setup, Rel$^{NTD}$ (10 μM in 50 mM Tris-HCl, pH 8.3, 200 mM NaCl) was placed in the sample cell, and DarB (100 μM in the same buffer) was placed in the titration syringe. All experiments were carried out at 20 °C with and a stirring speed of 307 rpm. Data analysis was carried out using MicroCal PEQ-ITC Analysis, Malvern Panalytical software.

**Bacterial two-hybrid assay**. Primary protein–protein interactions were identified by bacterial two-hybrid (BACTH) analysis[53]. The BACTH system is based on the interaction-mediated reconstruction of *Bordetella pertussis* adenylate cyclase (CyaA) activity in *E. coli* BTH101. Functional complementation between two fragments (T18 and T25) of CyaA as a consequence of the interaction between bait and prey molecules results in the synthesis of cAMP, which is monitored by measuring the β-galactosidase activity of the cAMP-CAP-dependent promoter of the *E. coli lac* operon. Plasmids pUT18C and p25N allow the expression of proteins fused to the T18 and T25 fragments of CyaA, respectively. For these experiments, we used the plasmids pGP2974-pGP2977, which encode N-and C-terminal fusions of T18 or T25 to *darB*. Accordingly, plasmids pGP2982-pGP2985 were used for *ccpC*, pGP3344-pGP3347 for *rel*, pGP3415-pGP3418 for *rel*(SYN-RRM), pGP3419-pGP3422 for *rel*(Rel$^{NTD}$), pGP3336-pGP3339 for *sasA*, and pGP3411-pGP3414 for *sasB*. These plasmids were used for co-transformation of *E. coli* BTH101 and the protein–protein interactions were then analyzed by plating the cells on LB plates containing 100 μg/ml ampicillin, 50 μg/ml kanamycin, 40 μg/ml X-Gal (5-bromo-4-chloro-3-indolyl-ß-D-galactopyranoside), and 0.5 mM IPTG (isopropyl-ß-D-thiogalactopyranoside). The plates were incubated for a maximum of 36 h at 28 °C.

**Total RNA preparation**. For RNA isolation, the cells were grown in MSSM minimal medium with 0.1 mM KCl to an OD$_{600}$ of 0.4–0.6 and harvested. Preparation of total RNA was carried out using the RNeasy Plus Mini Kit (Qiagen, Hilden, Germany)[59]. RNA was visualized using a 1% agarose formaldehyde gel in MOPS buffer (20 mM MOPS, 5 mM Na-Acetate, 1 mM EDTA, pH 7). The gel was stained with ethidium bromide.

**Quantification of (p)ppGpp in *B. subtilis* cell extracts**. Bacteria were precultured in LB medium and in MSSM medium with 0.1 mM KCl, and this preculture was used to inoculate the main culture in MSSM medium with 0.1 mM KCl. The cultures were grown until the exponential phase. The nucleotides were extracted and quantified by the SPE Extraction (modified from[60]). Briefly, 2 ml of the culture were mixed with 75 μl 100% formic acid and incubated on ice for 30 min. After addition of 2 ml 50 mM ammonium acetate (pH 4.5), precipitates were removed by centrifugation at 3000 × *g* for 5 min. The supernatant was transferred onto a pre-washed SPE column (OASIS Wax cartridges 1 cc, Waters). Prewashing was done with 1 ml methanol (4500 × *g* for 1 min) and 1 ml 50 mM ammonium acetate (pH 4.5) (3800 × *g* for 1 min). The supernatant was loaded in 1 ml steps (4 times) by 1 min centrifugation at 3800 × *g* each time. The SPE column was washed with 1 ml 50 mM ammonium acetate (pH 4.5, 3800 × *g* for 1 min) and 1 ml methanol (3800 × *g* for 1 min). After elution with 1 ml 80% ddH$_2$O, 20% methanol, 3% NH$_4$OH into a new tube and centrifugation (3800 × *g* for 1 min), the samples were frozen in liquid nitrogen and freeze-dried. The nucleotides were analyzed by liquid chromatography coupled tandem mass spectrometry on a QTRAP 5500 instrument (Sciex, Framingham, Massachusetts) equipped with an electrospray ionization source (ESI). Data were recorded in the multiple reaction monitoring (MRM) mode. Separation was performed on a Hypercarb column (30 × 4.6 mm, 5 μm particle size; Phenomenex, Aschaffenburg, Germany) using a linear gradient of solvent A (10 mM ammonium acetate pH 10) and solvent B (acetonitrile) at a flow rate of 0.6 ml/min, with solvent B with a gradient of 4–60% B being delivered within 8 minutes. The ppGpp and pppGpp standards were purchased from Jena Bioscience.

**Rel synthetase activity assay**. Rel, and the wild type and A25G-R132M mutant DarB proteins were purified as described above. The assay was carried out in HEPES:Polymix buffer (20 mM HEPES:KOH pH 7.5, 2 mM DTT, 5 mM Mg (OAc)$_2$, 95 mM KCl, 5 mM NH$_4$Cl, 0.5 mM CaCl$_2$, 8 mM putrescine, 1 mM spermidine)[21]. The activity of Rel was measured alone or in the presence of DarB, or the control protein BSA. The assay was carried out at 37 °C and the reaction was started by addition of 1 mM ATP and 1 mM GTP (Jena Bioscience), and samples for the nucleotide measurement were taken after 15 min. The nucleotides were extracted and quantified by SPE Extraction modified from[60] as described above, with the exception that 500 μl of the assay mix was mixed with 1500 μl assay buffer and with 75 μl 100% formic acid and incubated on ice for 30 min.

**Rel hydrolase activity assay**. The pppGpp hydrolysis assay was carried out in HEPES:Polymix buffer (20 mM HEPES:KOH pH 7.5, 2 mM DTT, 5 mM Mg (OAc)$_2$, 95 mM KCl, 5 mM NH$_4$Cl, 0.5 mM CaCl$_2$, 8 mM putrescine, 1 mM spermidine), 1 mM MnCl$_2$)[21]. The activity of Rel was measured alone or in the presence of DarB, or the control protein BSA. The assay was carried out at 37 °C

and the reaction was started by addition of 1 mM pppGpp (Jena Bioscience), and samples for the nucleotide measurement were taken after 15 min. The nucleotides were extracted and quantified as described above.

**Reporting summary**. Further information on research design is available in the Nature Research Reporting Summary linked to this article.

## Data availability

The mass spectrometry proteomics data have been deposited to the ProteomeXchange Consortium via the PRIDE[61] partner repository with the dataset identifier PXD018087. All the data of this study are available within the paper or can be requested from the corresponding author. Source data are provided with this paper.

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

## Acknowledgements

We wish to thank Vasili Hauryliuk for providing an expression vector and a detailed protocol for the purification of *B. subtilis* Rel. We are grateful to Oliver Valerius for the help with LCMS analyses which were done at the Service Unit LCMS Protein Analytics of the Göttingen Center for Molecular Biosciences (GZMB) at the Georg-August-University Göttingen (Grant ZUK 41/1 DFG-GZ A 630 to G.H. Braus and grant DFG-GZ: INST 186/1230-1 FUGG to S. Pöggeler). We are grateful to Daniel Zeigler (Bacillus Genetic Stock Center) and Gert Bange for providing *B. subtilis* Rel mutant strains. We wish to thank Gabriele Beyer, Mats Koschel, and Tobias Krammer for helpful discussions and technical assistance. Annette Garbe is acknowledged for the nucleotide analysis. This work was supported by grants of the Deutsche Forschungsgemeinschaft (DFG) within the Priority Program SPP1879 (to R.F. and J.S.) and INST186/1117 (to R.F.).

## Author contributions

L.K., R.F., and J.S. conceptualized the study. L.K., C.H., D.W., H.B., J.L.H., A.D., and K.S. developed the methodology, performed the experiments and analyzed the data. L.K. and J.S. wrote the original draft of the manuscript. J.L.H., A.D., K.S., and R.F. reviewed and edited the manuscript. R.F. and J.S. acquired funding. R.F., A.D., and J.S. provided supervision.

## Funding

## Competing interests

The authors declare no competing interests.
