## [Peer Review File · Nature Communications]

REVIEWER COMMENTS

Reviewer #1 (Remarks to the Author):

Krüger and colleagues investigate the interplay between two *Bacillus subtilis* sensory/regulatory proteins: bifunctional RelA-SpoT Homolog enzyme Rel which produces and hydrolyses alarmone nucleotide (p)ppGpp and c-di-AMP binder DarB. This is, if I am not mistaken, the first paper from the Stülke lab that deals with Rel, which is probably the reason for the ‘teething’ problems. The authors convincingly demonstrate the interaction between the two proteins using both microbiological and biochemical assays. However, I am less convinced about the functional assays, as detailed below. However, before we dive into these, some comments about RSH terminology and references:

RSH terminology and references, 1. Most of bacteria encode one long bifunctional RSH enzyme, Rel (see Mittenhuber, *J Mol Microbiol Biotechnol.* 2001 Oct;3(4):585-600, followed by Aktinson *PLOS ONE* 2011, Hauryliuk et al. *Nat Rev Micro* 2015). On the lineage to beta and gammaproteobacteria rel gene has duplicated and diversified into a pair of RelA and SpoT. Importantly, RelA is a synthetase (SYNTH) active, hydrolase (HD) inactive enzyme. Krüger and colleagues refer to *B. subtilis* long RSH as RelA (for historic reasons... but this does not make it less confusing) and specify only the SYNTH activity the first time the enzyme is discussed: “RelA catalyzes the production of the alarmones ppGpp and pppGpp by transferring pyrophosphate derived from ATP to GDP and GTP, respectively, under conditions of amino acid starvation⁴.” This is highly misleading, because while, yes, RelA only has SYNTH activity, Rel has both HD and SYNTH. Note that the reference used here is not very relevant (Steinchen, W. & Bange, G. The magic dance of the alarmones (p)ppGpp.): it is a review focusing on the molecular mechanisms of (p)ppGpp-mediated regulation of the target proteins. I recommend citing a relevant original paper that would describe / characterise the enzymatic activities of *B. subtilis* Rel. To my knowledge (which is far from exhaustive...), the only dedicated biochemical study directly following the enzymatic activities of this factor is Takada et al. 2020 *Frontiers Microbiology*.

RSH terminology and references, 2. There are several RSH domain terminologies out there, which is confusing. The reason being is that several cryo-EM papers came out basically simultaneously, and they have identified (or renamed based on the structural observations) new domains that were missed by previous studies relying purely in silico domain annotation. Krüger and colleagues use a modified version of one of the existing terminologies. I suggest the following modifications:

- specify which of the nomenclatures you are following and cite the original paper when you introduce it – is it Brown et al.?

- NTD and CTD are not domains, they are domain regions (since they contain several domains themselves; a domain cannot contain domains)

- line 525 'ZFD, a putative zinc 525 finger domain' – there is nothing putative about ZFD: it looks like a ZFD (interacts with the A-site rRNA finger element as per cryo-EM papers (Brown, Loveland and Arenz)) and acts as a ZFD (the contact is functional and stabilises RSH on the ribosome, Kudrin NAR 2019).

RSH terminology and references, 3. Please use Rel^{NTD} (NTD could be superscript or subscript) instead of just Rel when relevant: sometimes it becomes unclear if the experiment is performed with full-length or NTD-only version of the protein (e.g. Fig 2d table lists Rel Mw, while it is Rel^{NTD}).

Now, back to the data.

As I say, the authors convincingly show the interaction between Rel and DarB. That said, Rel is an exceedingly treacherous protein. It has issues with solubility and is hard to purify from contaminating ribosomes (see Takada et al. 2020 *Frontiers Microbiology*). However, the NTD region alone is much more soluble than the full-length protein and does not come contaminated with ribosomes (this one of the reasons why it is a must specifying when the NTD is used – and when the full-length is used, these two are very different beasts). The authors localise the binding to Rel^{NTD}, which suggests that the interaction could be compatible with Rel binding to the ribosome. The ribosome, or, rather, the 'starved' complex, is the key regulator of Rel. By itself it has a predominantly (p)ppGpp hydrolysis activity (see Takada et al. 2020 *Frontiers Microbiology*), with is dependent, as in the case of all HD-active RSH, on Mn²⁺ ions.

Which brings us to the functional studies, the main selling point of the paper, 'Fig. 3. DarB is an activator of pppGpp synthesis by RelA.' The biochemical assays were performed in the presence of Mn²⁺, therefore HD and SYNTH were not uncoupled, allowing simultaneous synthesis and degradation of pppGpp. The turnovers are presented in pmol pppGpp per ug of Rel. Enzymatic data should be presented in conversions per concentration per time, not per ug of protein. If one would do this simple math, it is clear that the activity is extremely low.

- It is impossible to determine if the effect of DarB (if there is an effect) is on SYNTH or HD of Rel. Biochemical experiments should assess both enzymatic activities. Please present the turnovers per molecule or uM or enzyme so it is possible to compare to other biochemical investigations of *B. subtilis* Rel (that is Takada et al. 2020 *Frontiers Microbiology*). Adding or omitting Mn²⁺ allows deconvoluting the two enzymatic activities.

- How relevant are the effects? In the cell Rel is regulated by the ribosome, which modulates the activity dramatically, suppressing the HD and activating the SYNTH. Would make sense to add the ribosome to the mix, but I guess it is hard.

In the current version biochemical assays show that the enzyme is basically inactive and becomes a bit less inactive. However, the observed activity is, numerically speaking, not physiologically relevant.

Crucial: the authors suggest that c-di-AMP is reactivating the interaction between Rel and DarB. This is an excellent opportunity to test if the effect is specific: addition of c-di-AMP should abrogate the effect of DarB on Rel in enzymatic assays. Given that no point mutants disrupting the interaction were selected genetically to probe the specificity of the B2H assays (I will get to this point later), this control would be very useful.

With the biochemical assays being challenging, one could just follow the activity in the cell by directly measuring the concentrations of ppGpp and pppGpp using ³²P TCL (easy and dirty) or HPLC (less easy - but quantitative). This is the most logical experiment given the title of the figure and is an absolute must-have. Ideally one should measure both c-di-AMP (LC-MS?) and (p)ppGpp in a set of mutant strains challenged by low potassium to directly probe the model proposed by the authors (Fig 4.)

Therefore, given the weakness of the functional assays – the authors mostly use indirect assays (growth and mild rRNA processing defects) while direct assays (nucleotide measurements) are available – I feel that the model (Fig. 4) is exceedingly premature. Functional assays need beefing up a lot before one could propose a model.

That said, the binding studies are convincing, and I suggest re-visiting these get some more ammo before plunging into the functional studies.

- The authors have their microbiological assays set up, so screening for point mutants that disrupt the interaction is a good start for teasing the system apart.
- The authors have purified the full-length Rel protein, but do the binding studies using SEC chromatography with RelNTD (labelled as RelA; please amend, it is not a full-length protein) – why? You have the materials, please use the full-length. Note that Takada et al. 2020 Frontiers Microbiology had some issues purifying RNA-free Rel in one step, please compare your protocol with theirs. Please report your 260/280 ratios for pure proteins, it is very informative while working with RNA binders.
- Judging by the SEC, Rel:DarB complex is very tight. Please titrate DarB to establish the stoichiometry: using apparent Mw to establish the stoichiometry is not a reliable approach (or just simply tone down the stoichiometry statement, this is not really the key finding here...).

To conclude. This is an interesting study identifying an interaction between Rel and DarB in *B. subtilis*. However, the project is currently very far from establishing the significance of this interaction. Direct measurements of (p)ppGpp and proper biochemistry following both SYNTH and HD are a must. Selecting specific point mutants would be a great way to strengthen the specificity of the functional assays, but is not a must: the authors should take advantage of the proposed effect of c-di-AMP on DarB:Rel complex formation in their enzymatic assays. Finally, tightening the RSH nomenclature / references (please cite original papers instead of reviews when possible, especially given that often the focus of the reviews does not coincide with the statement being made...) would improve the readability.

Reviewer #2 (Remarks to the Author):

The manuscript by Krüger et al. from Jörg Stülke's laboratory is a continuation of their analysis of the c-di-AMP-binding protein DarB. Interestingly, the secondary messenger c-di-AMP functions in K⁺ homeostasis of *B. subtilis* and similar organisms. In the present work, they show convincingly that the apo form of DarB interacts with the N-terminal domain of RelA *in vivo* and *in vitro* and stimulates pppGpp synthesis while the c-di-AMP-bound form of DarB has no such effect. Their data support a coherent physiological model explaining how the "rendezvous" of two secondary messengers control K⁺ homeostasis in *B. subtilis*. The manuscript presents a novel concept of metabolic control, the claims are novel, the data are solid and the conclusions justified.

A note on nomenclature is warranted. *E. coli* has two multidomain (p)ppGpp synthetases/hydrolases, RelA and SpoT, while *B. subtilis* has one, inconsistently called Rel (PMID: 32184768; PMID: 21858139; PMID: 27875634; PMID: 32176689), RelA (PMID: 25899641; PMID: 25331430; PMID: 24163341; PMID: 21709092) or RSH (PMID: 24065631), etc. A brief sentence could clarify the issue. The RelA nomenclature as used here may be preferred over Rel as it is now known that RelA of *B. subtilis* binds to the ribosome much the same way as RelA of *E. coli*.

Below are suggestions for improvements of the manuscript, only some of which are demanding.

Lines 31-32 in the abstract: this is a blunt statement and should be modified. For example, Germain et al., *Nat Commun* 2019 10(1):5763 showed that YtfK of *E. coli* stimulates ribosome-independent ppGpp synthesis of the multidomain RelA-homologue SpoT. This is important in the present context

because RelA (or Rel) of *B. subtilis* is also a SpoT homologue. The article by Germain et al. 2019 should be cited in the present manuscript.

46: Please define “Bateman domain”.

59: It is not possible for this referee to understand Supplementary Table 1 because the column heading terms are not defined. Please amend.

61-62: It should be mentioned that RelA of *B. subtilis* (similar to SpoT but dissimilar to RelA of *E. coli*) also hydrolyzes pppGpp and ppGpp.

65: “ β -lactamase” should be “ β -galactosidase”

80: Please consider that “is very closely related to” should/could be “function in”

105: “Thus, the interaction of DarB is specific for RelA.” should be “Thus, the interaction of DarB is most likely specific for RelA.”

111: Rather than referring to a review it might be warranted to refer to the original articles showing that the C-terminal domains of RelA mediate the binding to the ribosome.

123: Please define “SEC-MALS” at first use in text.

125: Please delete “excellent”, it is just cosmetics

155-162: Using the pre-rRNA levels as a read-out of ppGpp levels is indirect and Fig 3c should probably be moved to Extended data (or even deleted). The direct measurement of (p)ppGpp levels would be much clearer. If this is not possible it should be explained why.

167: Please include an original study supporting the statement.

163-176 and Figure 3d: If possible, please show the original data revealing that apo-DarB stimulates the synthesis of pppGpp by RelA in Extended data.

Strengthening of the claims:

The authors could consider generating mutants of DarB that reduce the interaction with RelA, useful as specificity-controls in the in vitro assay (Figure 3d) and further analysis of the mechanism of activation. This possibility represents some additional work and is not demanding but would increase the strength of this central issue.

Reviewer #3 (Remarks to the Author):

This manuscript entitled “A rendezvous of two second messengers: The c-di-AMP receptor protein DarB controls (p)ppGpp synthesis in *Bacillus subtilis*” by Krüger et al describe the functional role of a hypothetical protein DarB. The authors have used microbiology and biochemistry approaches to provide evidence for the role of DraB in c-di-AMP mediated activation of RelA. However, this study has following concerns at the experimental level and the writing style of the manuscript.

1. The authors have used two constructs of RelA protein from *Bacillus subtilis*, one is full-length RelA (RelA) and another one is the hydrolase-synthetase domain of RelA (RelA-HS) but there is no consistency in the text and figures labeling while describing these proteins. It would be appropriate that specific terms RelA and RelA-HS should be used for clear understanding.
2. It has been reported earlier (Takada et al *Front Microbiol.* 2020; 11: 277) that RelA co-elutes with RNA contamination during size exclusion chromatography. Therefore, it would be appropriate here to include absorbance of the purified sample at 260 nm or 254 nm so that the RNA contamination issue can be ruled out.

3. It should be clearly described in the figure and method section that SEC-MALS experiments were carried out with full-length RelA and/or RelA HS (hydrolase+synthetase domain). Also, it should be mentioned that results were obtained by S75 and/or S200 column.
4. For the SEC-MALS experiments, the ratio of RelA and DraB used as 1:1 ratio in method section while in the figure legends, DraB is used three-fold molar ratio instead of equimolar in the SEC-MALS experiments and finally interactions were observed as 2:2 stoichiometry. Which stoichiometry would be appropriate for this interaction?
5. The bacterial two-hybrid system reported by the authors activates the synthesis of cAMP by adenylate cyclase. cAMP shares structural similarity with c-di-AMP and c-di-AMP to prevent the interaction of DraB and RelA. The authors should discuss that cAMP synthesis in an assay does not affect in vivo interaction studies between DraB and RelA.
6. Page 3 line 65-66, it is reported that RelA exhibit self-interaction in SEC-MALS while it is shown in the figure 2d that oligomerization status of RelA is a contradictory one. The confusion is further deepened by the detection of peaks at unpredicted volume. Additionally, the authors cited two references to support self-interaction results. First of all these two references are not based on structural studies as mentioned in the manuscript. Moreover, self-interactions reported in these references are the due presence of the C-terminal domain which responsible for oligomerization. While this study employed a construct that lacks C-terminal domains for self-interaction experiments and therefore, it may not be in agreement with cited studies.
7. Page 8 and line no 195 describes huge body of evidence for RelA is triggered by uncharged tRNA at the A site of ribosome. However, there is no experimental evidence for the presence of uncharged tRNA on the A site of ribosome prior binding of RelA. The uncharged tRNA mediated activation of RelA is still not clear. There are evidences of RelA binding to the uncharged tRNA before loading to the ribosome (Winther et al Mol Cell, 2018, 70, 95–105 ; Kushwaha et al. Curr Genet 2019, 65,1173-1184).
8. It reported previously that the growth of RelA deleted strain in minimal media is reduced compared to wild type strain. Therefore, amino acid must be supplemented in minimal media to compare the growth curve. The authors should discuss this point in the manuscript reporting growth curve of RelA deleted strain.
9. The authors must perform biophysical assays such as ITC, SPR, BLI, MST that would certainly provide quantitative evidence for these the interactions regulated by c-di-AMP.

Rebuttal

General

We are grateful to the reviewers who provided sometimes really demanding and challenging but always constructive and helpful comments. In the revised manuscript, we include a substantial amount of novel experimental data to meet the expectations of the reviewers. Specifically these include:

- Use of an improved construct and protocol for Rel purification
- *In vitro* analysis of interaction also with the full-length Rel protein
- Construction and characterization of a DarB mutant protein that is deficient in the interaction with Rel, this protein/ allele is included in most of the experimental setups
- New data to show that DarB affects both the synthetase and hydrolase activities *in vitro*, and that this is only the case in the absence of c-di-AMP
- Determination of (p)ppGpp concentrations *in vivo*

Reviewer #1

Krüger and colleagues investigate the interplay between two *Bacillus subtilis* sensory/regulatory proteins: bifunctional RelA-SpoT Homolog enzyme Rel which produces and hydrolyses alarmone nucleotide (p)ppGpp and c-di-AMP binder DarB. This is, if I am not mistaken, the first paper from the Stülke lab that deals with Rel, which is probably the reason for the ‘teething’ problems. The authors convincingly demonstrate the interaction between the two proteins using both microbiological and biochemical assays. However, I am less convinced about the functional assays, as detailed below. However, before we dive into these, some comments about RSH terminology and references:

RSH terminology and references, 1. Most of bacteria encode one long bifunctional RSH enzyme, Rel (see Mittenhuber, *J Mol Microbiol Biotechnol.* 2001 Oct;3(4):585-600, followed by Aktinson *PLOS ONE* 2011, Haurlyuk et al. *Nat Rev Micro* 2015). On the lineage to beta and gammaproteobacteria rel gene has duplicated and diversified into a pair of RelA and SpoT. Importantly, RelA is a synthetase (SYNTH) active, hydrolase (HD) inactive enzyme. Krüger and colleagues refer to *B. subtilis* long RSH as RelA (for historic reasons... but this does not make it less confusing) and specify only the SYNTH activity the first time the enzyme is discussed: “RelA catalyzes the production of the alarmones ppGpp and pppGpp by transferring pyrophosphate derived from ATP to GDP and GTP, respectively,

under conditions of amino acid starvation⁴. This is highly misleading, because while, yes, RelA only has SYNTH activity, Rel has both HD and SYNTH. Note that the reference used here is not very relevant (Steinchen, W. & Bange, G. The magic dance of the alarmones (p)ppGpp.): it is a review focusing on the molecular mechanisms of (p)ppGpp-mediated regulation of the target proteins. I recommend citing a relevant original paper that would describe / characterise the enzymatic activities of *B. subtilis* Rel. To my knowledge (which is far from exhaustive...), the only dedicated biochemical study directly following the enzymatic activities of this factor is Takada et al. 2020 *Frontiers Microbiology*.

Response: We see the point and the potential confusion caused by calling the long bifunctional *Bacillus* enzyme RelA. On the other hand, this designation is used in most of the literature on this gene/ protein. Judging both points, we decided to give avoiding confusion the priority, and do now use the Rel nomenclature. We also have replaced the Steinchen reference by the Takada paper as suggested by the reviewer.

RSH terminology and references, 2. There are several RSH domain terminologies out there, which is confusing. The reason being is that several cryo-EM papers came out basically simultaneously, and they have identified (or renamed based on the structural observations) new domains that were missed by previous studies relying purely in silico domain annotation. Krüger and colleagues use a modified version of one of the existing terminologies. I suggest the following modifications:

- specify which of the nomenclatures you are following and cite the original paper when you introduce it – is it Brown et al.?
- NTD and CTD are not domains, they are domain regions (since they contain several domains themselves; a domain cannot contain domains)
- line 525 'ZFD, a putative zinc 525 finger domain' – there is nothing putative about ZFD: it looks like a ZFD (interacts with the A-site rRNA finger element as per cryo-EM papers (Brown, Loveland and Arenz)) and acts as a ZFD (the contact is functional and stabilises RSH on the ribosome, Kudrin NAR 2019).

Response:

- **We do now throughout the paper adhere to the domain nomenclature as proposed by Brown et al., this is also cited.**
- **We agree, and use the terms N- (or C-) terminal parts or regions now**
- **We have deleted “putative” for the zinc finger domain**

RSH terminology and references, 3. Please use Rel^{NTD} (NTD could be superscript or subscript) instead of just Rel when relevant: sometimes it becomes unclear if the experiment is performed with full-length or NTD-only version of the protein (e.g. Fig 2d table lists Rel Mw, while it is Rel^{NTD}).

Response: This has been corrected. We do now use Rel^{NTD} for the N-terminal part of the protein throughout the ms.

Now, back to the data.

As I say, the authors convincingly show the interaction between Rel and DarB. That said, Rel is an exceedingly treacherous protein. It has issues with solubility and is hard to purify from contaminating ribosomes (see Takada et al. 2020 Frontiers Microbiology). However, the NTD region alone is much more soluble than the full-length protein and does not come contaminated with ribosomes (this one of the reasons why it is a must specifying when the NTD is used – and when the full-length is used, these two are very different beasts). The authors localise the binding to Rel^{NTD}, which suggests that the interaction could be compatible with Rel binding to the ribosome. The ribosome, or, rather, the ‘starved’ complex, is the key regulator of Rel. By itself it has a predominantly (p)ppGpp hydrolysis activity (see Takada et al. 2020 Frontiers Microbiology), with is dependent, as in the case of all HD-active RSH, on Mn²⁺ ions.

Which brings us to the functional studies, the main selling point of the paper, ‘Fig. 3. DarB is an activator of pppGpp synthesis by RelA.’ The biochemical assays were performed in the presence of Mn²⁺, therefore HD and SYNTH were not uncoupled, allowing simultaneous synthesis and degradation of pppGpp. The turnovers are presented in pmol pppGpp per ug of Rel. Enzymatic data should be presented in conversions per concentration per time, not per ug of protein. If one would do this simple math, it is clear that the activity is extremely low.

Response: We are grateful to the reviewer for drawing our attention to this issue of metal ions. We have repeated the analyses exactly as described by Takada et al. The new results are shown in the revised manuscript (see Fig. 4a and 4b).

- It is impossible to determine if the effect of DarB (if there is an effect) is on SYNTH or HD of Rel. Biochemical experiments should assess both enzymatic activities. Please present the turnovers per molecule or uM or enzyme so it is possible to compare to other biochemical investigations of B.

subtilis Rel (that is Takada et al. 2020 Frontiers Microbiology). Adding or omitting Mn²⁺ allows deconvoluting the two enzymatic activities.

Response: In the revised manuscript, we report data for both activities and we also provide turnover. The new results are shown in the revised manuscript (see Fig. 4a and 4b).

- How relevant are the effects? In the cell Rel is regulated by the ribosome, which modulates the activity dramatically, suppressing the HD and activating the SYNTH. Would make sense to add the ribosome to the mix, but I guess it is hard.

Response: Based on the data of our DarB overexpression experiment (reduced growth and suppression by Rel deletion), we feel that the effect is physiologically relevant. For the biochemical assays, we did not add ribosomes to be able to see the clear effect of DarB.

In the current version biochemical assays show that the enzyme is basically inactive and becomes a bit less inactive. However, the observed activity is, numerically speaking, not physiologically relevant.

Response: We agree. Fortunately, we got an expression construct and protocol from one of the experts in the field, and with this the purification went much better. Moreover, the measured activities are now much higher (see Fig. 4a and 4b).

Crucial: the authors suggest that c-di-AMP is reactivating the interaction between Rel and DarB. This is an excellent opportunity to test if the effect is specific: addition of c-di-AMP should abrogate the effect of DarB on Rel in enzymatic assays. Given that no point mutants disrupting the interaction were selected genetically to probe the specificity of the B2H assays (I will get to this point later), this control would be very useful.

Response: We agree. Data with c-di-AMP addition are included in the revised version (see Fig. 4a and 4b).

With the biochemical assays being challenging, one could just follow the activity in the cell by directly measuring the concentrations of ppGpp and pppGpp using ³²P TCL (easy and dirty) or HPLC (less easy - but quantitative). This is the most logical experiment given the title of the figure and is an absolute must-have. Ideally one should measure both c-di-AMP (LC-MS?) and (p)ppGpp in a set of

mutant strains challenged by low potassium to directly probe the model proposed by the authors (Fig 4.)

Response: Fortunately, we could collect all the necessary in vitro evidence for our model. On the other hand, we also tried assaying the nucleotides, and we found that this was the much more challenging problem. If we have low potassium concentrations in the medium, c-di-AMP is still being made (see Gundlach et al, 2017). To get to no c-di-AMP, we would need a potassium starvation that prevents growth. Therefore, we performed analysis of the nucleotide pools under the conditions that correspond to the growth defect (DarB overexpression vs. wild type and darB mutant). These data are included (Fig. 4c), and they are in excellent agreement with our model.

Therefore, given the weakness of the functional assays – the authors mostly use indirect assays (growth and mild rRNA processing defects) while direct assays (nucleotide measurements) are available – I feel that the model (Fig. 4) is exceedingly premature. Functional assays need beefing up a lot before one could propose a model.

Response: We are sure that all the new data that we have added in the revised version will convince the reviewer that our model is correct.

That said, the binding studies are convincing, and I suggest re-visiting these get some more ammo before plunging into the functional studies.

- The authors have their microbiological assays set up, so screening for point mutants that disrupt the interaction is a good start for teasing the system apart.

Response: Actually, getting meaningful point mutations that disrupt the interaction is not that easy as many mutations may interfere with the general folding of the protein. However, based on the structure of the DarB-c-di-AMP complex (Heidemann and Ficner, unpublished), we have chosen two amino acids (A25, R132) that could be replaced individually and in combination. As shown in the revised manuscript, the double mutant binds to Rel only weakly. We are certain that this protein folds correctly since it can still bind c-di-AMP (see Supplementary Figure 4).

- The authors have purified the full-length Rel protein, but do the binding studies using SEC chromatography with ReINTD (labelled as RelA; please amend, it is not a full-length protein) – why? You have the materials, please use the full-length. Note that Takada et al. 2020 Frontiers Microbiology had some issues purifying RNA-free Rel in one step, please compare your protocol with

theirs. Please report your 260/280 ratios for pure proteins, it is very informative while working with RNA binders.

Response: As mentioned above, we have used a new superior Rel expression construct and protocol, and have performed SEC-MALS also with full-length Rel (see Supplementary Figure 3). For purity, we have performed extensive washes with 4 M NaCl. Moreover, the final protein was obtained after SEC. To demonstrate the purity, we have performed a protein gel as well as an agarose gel to verify that the protein is free of RNA (see Supplementary Figure 2). Moreover, we also state the 260/280 ratio as. This is 0.9 and can be found in the legend to Supplementary Figure 2.

- Judging by the SEC, Rel:DarB complex is very tight. Please titrate DarB to establish the stoichiometry: using apparent Mw to establish the stoichiometry is not a reliable approach (or just simply tone down the stoichiometry statement, this is not really the key finding here...).

Response: To determine kinetic parameters, we have now performed ITC (see Fig. 2e).

To conclude. This is an interesting study identifying an interaction between Rel and DarB in *B. subtilis*. However, the project is currently very far from establishing the significance of this interaction. Direct measurements of (p)ppGpp and proper biochemistry following both SYNTH and HD are a must. Selecting specific point mutants would be a great way to strengthen the specificity of the functional assays, but is not a must: the authors should take advantage of the proposed effect of c-di-AMP on DarB:Rel complex formation in their enzymatic assays. Finally, tightening the RSH nomenclature / references (please cite original papers instead of reviews when possible, especially given that often the focus of the reviews does not coincide with the statement being made...) would improve the readability.

Response: We are really grateful for the challenging but in principle positive comments by this reviewer. By performing nearly all experiments that he/she suggested, we hope we can now convince him/her that our conclusions and our model are correct.

Reviewer #2 (Remarks to the Author):

The manuscript by Krüger et al. from Jörg Stülke's laboratory is a continuation of their analysis of the c-di-AMP-binding protein DarB. Interestingly, the secondary messenger c-di-AMP functions in K+

homeostasis of *B. subtilis* and similar organisms. In the present work, they show convincingly that the apo form of DarB interacts with the N-terminal domain of RelA in vivo and in vitro and stimulates pppGpp synthesis while the c-di-AMP-bound form of DarB has no such effect. Their data support a coherent physiological model explaining how the “rendezvous” of two secondary messengers control K⁺ homeostasis in *B. subtilis*. The manuscript presents a novel concept of metabolic control, the claims are novel, the data are solid and the conclusions justified.

Response: We are very grateful to this reviewer for these kind words about our work.

A note on nomenclature is warranted. *E. coli* has two multidomain (p)ppGpp synthetases/hydrolases, RelA and SpoT, while *B. subtilis* has one, inconsistently called Rel (PMID: 32184768; PMID: 21858139; PMID: 27875634; PMID: 32176689), RelA (PMID: 25899641; PMID: 25331430; PMID: 24163341; PMID: 21709092) or RSH (PMID: 24065631), etc. A brief sentence could clarify the issue. The RelA nomenclature as used here may be preferred over Rel as it is now known that RelA of *B. subtilis* binds to the ribosome much the same way as RelA of *E. coli*.

Response: Based on the comments of reviewer #1, we do now use the term Rel throughout the manuscript.

Below are suggestions for improvements of the manuscript, only some of which are demanding.

Lines 31-32 in the abstract: this is a blunt statement and should be modified. For example, Germain et al., Nat Commun 2019 10(1):5763 showed that YtfK of *E. coli* stimulates ribosome-independent ppGpp synthesis of the multidomain RelA-homologue SpoT. This is important in the present context because RelA (or Rel) of *B. subtilis* is also a SpoT homologue. The article by Germain et al. 2019 should be cited in the present manuscript.

Response: We have deleted the “first example claim” in the abstract. Moreover, we do now refer to the mentioned paper in the discussion, and the paper is cited (ref. 34).

46: Please define “Bateman domain”.

Response: This is now defined as follows: “consists of a tandem of two CBS (cystathionine-beta synthase) domains, an arrangement called Bateman domain”. Moreover, a reference is provided (ref. 13).

59: It is not possible for this referee to understand Supplementary Table 1 because the column heading terms are not defined. Please amend.

Response: The legends have been modified. They adhere to the standard of proteomics data presentation.

61-62: It should be mentioned that RelA of *B. subtilis* (similar to SpoT but dissimilar to RelA of *E. coli*) also hydrolyzes pppGpp and ppGpp.

Response: The corresponding statement has been added.

65: “ β -lactamase” should be “ β -galactosidase”

Response: Thank you for noting, corrected!

80: Please consider that “is very closely related to” should/could be “function in”

Response: this has been modified as suggested.

105: “Thus, the interaction of DarB is specific for RelA.” should be “Thus, the interaction of DarB is most likely specific for RelA.”

Response: this has been modified as suggested.

111: Rather than referring to a review it might be warranted to refer to the original articles showing that the C-terminal domains of RelA mediate the binding to the ribosome.

Response: We agree and cite now Brown et al., 2016 (ref. 19)

123: Please define “SEC-MALS” at first use in text.

Response: This is now defined: “we performed size exclusion chromatography-multiangle light scattering (SEC-MALS) experiments” (l. 101)

125: Please delete “excellent”, it is just cosmetics

Response: deleted as suggested.

155-162: Using the pre-rRNA levels as a read-out of ppGpp levels is indirect and Fig 3c should probably be moved to Extended data (or even deleted). The direct measurement of (p)ppGpp levels would be much clearer. If this is not possible it should be explained why.

Response: We agree and have moved the pre-rRNA levels to the Supplementary data (Supplementary Figure 9). Moreover, we have also included the direct (p)ppGpp measurements (see Fig. 4c).

167: Please include an original study supporting the statement.

Response: We agree and cite now Brown et al., 2016 (ref. 25)

163-176 and Figure 3d: If possible, please show the original data revealing that apo-DarB stimulates the synthesis of pppGpp by RelA in Extended data.

Response: The values are shown in the diagrams (Figs. 4a and 4b for hydrolysis).

Strengthening of the claims:

The authors could consider generating mutants of DarB that reduce the interaction with RelA, useful as specificity-controls in the in vitro assay (Figure 3d) and further analysis of the mechanism of activation. This possibility represents some additional work and is not demanding but would increase the strength of this central issue.

Response: Such a mutant has been generated and is now included in nearly all experiments.

Reviewer #3 (Remarks to the Author):

This manuscript entitled “A rendezvous of two second messengers: The c-di-AMP receptor protein DarB controls (p)ppGpp synthesis in *Bacillus subtilis*” by Krüger et al describe the functional role of a hypothetical protein DarB. The authors have used microbiology and biochemistry approaches to

provide evidence for the role of DraB in c-di-AMP mediated activation of RelA. However, this study has following concerns at the experimental level and the writing style of the manuscript.

1. The authors have used two constructs of RelA protein from *Bacillus subtilis*, one is full-length RelA (RelA) and another one is the hydrolase-synthetase domain of RelA (RelA-HS) but there is no consistency in the text and figures labeling while describing these proteins. It would be appropriate that specific terms RelA and RelA-HS should be used for clear understanding.

Response: We have taken care to label the constructs consistently throughout the manuscript. We do now use Rel^{NTD} for the N-terminal part of the protein throughout the ms.

2. It has been reported earlier (Takada et al Front Microbiol. 2020; 11: 277) that RelA co-elutes with RNA contamination during size exclusion chromatography. Therefore, it would be appropriate here to include absorbance of the purified sample at 260 nm or 254 nm so that the RNA contamination issue can be ruled out.

Response: We have now purified the Rel protein using the construct and the procedure described by Takada et al. To exclude the possibility of RNA contamination, we have performed an agarose gel (see Supplementary Figure 2). Moreover, we also state the 260/280 ratio as also requested by reviewer #1. This is 0.9 and can be found in the legend to Supplementary Figure 2.

3. It should be clearly described in the figure and method section that SEC-MALS experiments were carried out with full-length RelA and/or RelA HS (hydrolase+synthetase domain). Also, it should be mentioned that results were obtained by S75 and/or S200 column.

Response: It was a S200 column. This has been corrected.

4. For the SEC-MALS experiments, the ratio of RelA and DraB used as 1:1 ratio in method section while in the figure legends, DraB is used three-fold molar ratio instead of equimolar in the SEC-MALS experiments and finally interactions were observed as 2:2 stoichiometry. Which stoichiometry would be appropriate for this interaction?

Response: We have corrected the mistake in the figure legend. In the revised manuscript, the stoichiometry is based on the ITC experiment. It is 2:2 (DraB:Rel; two molecules of Rel bind to each one subunit of the DraB dimer). See also Fig. 5.

5. The bacterial two-hybrid system reported by the authors activates the synthesis of cAMP by adenylate cyclase. cAMP shares structural similarity with c-di-AMP and c-di-AMP to prevent the interaction of DraB and RelA. The authors should discuss that cAMP synthesis in an assay does not affect in vivo interaction studies between DraB and RelA.

Response: Actually, there is no significant structural similarity between c-di-AMP and cAMP. So, we don't expect any problem here. Moreover, we include more data to show that c-di-AMP interferes with the interaction between the two proteins (including Rel activity assays).

6. Page 3 line 65-66, it is reported that RelA exhibit self-interaction in SEC-MALS while it is shown in the figure 2d that oligomerization status of RelA is a contradictory one. The confusion is further deepened by the detection of peaks at unpredicted volume. Additionally, the authors cited two references to support self-interaction results. First of all these two references are not based on structural studies as mentioned in the manuscript. Moreover, self-interactions reported in these references are the due presence of the C-terminal domain which responsible for oligomerization. While this study employed a construct that lacks C-terminal domains for self-interaction experiments and therefore, it may not be in agreement with cited studies.

Response: We agree that this was not sufficiently clear. The self-interaction for Rel refers to the full-length protein, not to the NTD. With the consequent use of the distinct designations, this is now much clearer from the manuscript. We have also deleted the reference to the SEC-MALS experiment, and the reference is correct since the full-length protein with the CTD can oligomerize as also mentioned by the reviewer.

7. Page 8 and line no 195 describes huge body of evidence for RelA is triggered by uncharged tRNA at the A site of ribosome. However, there is no experimental evidence for the presence of uncharged tRNA on the A site of ribosome prior binding of RelA. The uncharged tRNA mediated activation of RelA is still not clear. There are evidences of RelA binding to the uncharged tRNA before loading to the ribosome (Winther et al Mol Cell, 2018, 70, 95–105 ; Kushwaha et al. Curr Genet 2019, 65,1173-1184).

Response: We now cite Brown et al., Nature, 2016 (ref. 25).

8. It reported previously that the growth of RelA deleted strain in minimal media is reduced

compared to wild type strain. Therefore, amino acid must be supplemented in minimal media to compare the growth curve. The authors should discuss this point in the manuscript reporting growth curve of RelA deleted strain.

Response: The *B. subtilis rel* mutant is known to have a growth defect in complex medium but not in minimal medium. This is in agreement with what we observe and show in Fig. 3a, 3b. The growth defect of the *rel* mutant in complex medium is caused by the accumulation of (p)ppGpp and the lack of the Rel degradation activity. Under our conditions in minimal medium, essentially no (p)ppGpp is formed in the *rel* mutant which is in perfect agreement with the absence of a growth defect.

9. The authors must perform biophysical assays such as ITC, SPR, BLI, MST that would certainly provide quantitative evidence for these the interactions regulated by c-di-AMP.

Response: As suggested by the reviewer, we have now included ITC experiments (see Fig. 2e, blue lines, apo-DarB; red lines, DarB-c-di-AMP complex).

REVIEWER COMMENTS

Reviewer #1 (Remarks to the Author):

As I have pointed out in the previous review, the authors convincingly argue that DarB can bind the NTD domain region of Rel, but both the mechanistic implications (i.e. what happens with (p)ppGpp synthesis and degradation by Rel) and the physiological significance of this interaction are not so clear. In the revised version, Krüger and colleagues did a great job at significantly beefing up the manuscript and expanding the methodology they use. They have put a significant effort into generating new data which would enhance the mechanistic aspect of the study. I understand that the authors are under certain level of time pressure since the results were already presented at the Subtillery 2020 conference, and I would guess that several labs are already actively working to validate the results.

That said, unfortunately, it feels that the authors have rushed the re-submission. There was no need: I am sure that there is no chance of scooping, given how small the *B. subtilis* stringent response community is. I am absolutely positive that we all want to see this paper published so we can build on these results. Since the effects reported in the revised version are modest or / and in direct conflict with a significant body of literature, an additional round of revision is, unfortunately, necessary.

Major concern: As Reviewer 3 has pointed out, Δ rel *B. subtilis* strain has a growth defect and requires amino acid supplements to offset this defect. The authors reply by saying that “rel mutant have a growth defect in complex medium but not in minimal medium. ... Under our conditions in minimal medium, essentially no (p)ppGpp is formed in the rel mutant which is in perfect agreement with the absence of a growth defect.”. This statement is very surprising and does not agree with the published data.

Several groups reported that in (p)ppGpp is essential for growth on minimal medium in the case of *E. coli* (Potrykus et al., 2011) and *B. subtilis* (Kriel et al., 2012). In the revised version, Krüger and colleagues report that there is no detectable pppGpp or ppGpp production in Δ rel mutant grown on minimal medium (Figure 4C). This is highly surprising: if the strain would have no (p)ppGpp, it would not grow on minimal medium at all. Δ rel *B. subtilis* should have (p)ppGpp levels much higher than the wild type: in the absence of Rel HD activity, (p)ppGpp synthesised by the SAS enzymes is not degraded, resulting in growth defect. One could argue that, somehow, in minimal media the situation is different. However, constative (p)ppGpp production was reported for *B. subtilis* Δ rel mutant grown on minimal medium in (Benoist et al., 2015). Similarly, both *S. mutants* (Lemos et al., 2007) and *E. faecalis* (Gaca et al., 2013) Δ rel mutants grow much slower in synthetic medium

compared with wild type due to accumulation of SAS-generated (p)ppGpp. Similarly, Δ rel B. subtilis engineered to express Rel NTD (which is virtually enzymatically inactive) under IPTG-inducible promoter was tested in growth assays on minimum and rich medium in the absence of IPTG (Takada et al., 2020). Under these conditions the strain is basically Δ rel, with no NTD expressed. As expected – and in direct contradiction with the statement from Krüger and colleagues – the growth defect of Δ rel B. subtilis was considerably more pronounced on minimal medium.

Essential experimental revision: Collectively, this suggests that something is seriously wrong with Δ rel B. subtilis used by Krüger and colleagues. This is not all that surprising. In our hands this strain is quite tricky to work because it picks up suppressor mutations (I would guess abrogating production of SAS), especially when grown on minimal medium. Since both HPLC and growth data do not fit the reported literature, I suggest requesting Δ rel B. subtilis from some other lab (I recommend the strain from Yuzuru Tozawa, it is well-characterised). The issues outlined above affect Figures 3B and 4C and should be addressed.

Essential text revision: Please provide a Supplementary table listing the strains used in the paper. Since the strain uses several strains (some of which were constructed specifically for this paper, including the potentially problematic Δ rel, see above), it is essential to document them.

Essential figure revision: In the revised version the authors report an HPLC analysis of nucleotide pools. These results crucial for assessing the physiological relevance of the Rel:DarB interaction. Since the authors use HPLC, they simultaneously measure several nucleotides (GTP / GDP; ATP / ADP; pppGpp / ppGpp). In the case of B. subtilis, the critical control parameter is not the concentration of (p)ppGpp; it is the ATP / GTP ratio (Krasny and Gourse, 2004). Since the authors already have the data, please plot the results as fractions of the nucleotide pools: the G pool, the A pool, the NTP pool; and please calculate the adenylate energy charge (AEC) so we can assess the quality of the data, it should be 0.8-0.95 (e.g. see (Varik et al., 2017)). No extra work is needed, all that is needed is re-plotting the data.

Essential experimental revision: However, what is needed – in order to put the results into perspective – is a positive control, i.e. acute stringent response induced by SHX / mupirocin. Right now, the data are presented in pmol x OD600 x ml⁻¹ of ppGpp / pppGpp, and it is hard to assess if the 2-fold increase is meaningful. The levels of (p)ppGpp vary dramatically across the growth curve (at least 20-fold (Varik et al., 2017)), so a mere 2-fold change could simply reflect the difference in growth rate/position on the growth curve, and not a specific effect of DarB on Rel SYNTH / HD. Stress caused by expressing a mildly toxic protein such as DarB could easily cause a 2-fold increase in (p)ppGpp.

Essential experimental revision: Since the effects are so weak (about 2x increase in (p)ppGpp) it is crucial to demonstrate that the growth defect caused by DarB overexpression is due to (p)ppGpp accumulation (direct causation) and not the opposite (growth defect caused by DarB toxicity is the cause of 2-fold (p)ppGpp increase). To do so, the authors should co-overexpress a hydrolase (e.g. MESH1 or SYNTH-inactive Rel) with DarB. This should restore the growth rate to wt (i.e. if we deal with causation, as the model implies). I notice that the authors have already received some plasmids from the Hauryliuk lab. I am sure the necessary materials for this experiment will be shared as well.

Minor revisions, largely addressable by rewriting / correcting the text and / or figures:

Model Figure 5: the model could (and should) be improved. Panel b displays DarB:Rel complex in the absence of c-di-AMP. Please integrate it into panel a (I would place it horizontally, the panel a) and show how c-di-AMP binds to DarB, dissociates the complex, show what happens with SYNTH / HD activates of Rel etc.; please connect the molecular model with the physiology one.

Biochemistry: the experimental procedures need to be described: what are the molar concentrations of Rel, DarB, c-di-AMP? What is the intracellular range of c-di-AMP concentrations (about 1-5 uM, right? Please provide the references). A specificity control (just c-di-AMP + Rel, no DarB) is missing in both SYNTH and HD activity test; c-di-AMP could be affecting Rel directly. It would be nice to have a titration of c-di-AMP and DarB. Note that the HD activity of your Rel is at least 10-fold lower that reported by Takada and colleagues, while you are using their construct and following (but not precisely, please see comment below) their purification procedure. Since the HD / SYNTH ratio is a key parameter for Rel (off the ribosome the factor is predominantly HD), please do mention this slight difference.

Please add the quantitative data to the figures, e.g. Kds on the ITC panels (Fig 2e, Kd is 0.65 uM) and growth rates on Figure 3abc.

ITC: please discuss the abundancy of Rel and DarB in the cell and contrast these with the Kd (0.65 uM is not very low; and this is in the absence of c-di-AMP; there is always some in the cell).

In the rebuttal the authors state: 'For the biochemical assays, we did not add ribosomes to be able to see the clear effect of DarB.' This is not a good argument. In the cell Rel is activated by the ribosomes even under normal growth: this is why deltaRRM mutant is toxic in a ribosome-dependant way under normal growth conditions. Rel has considerable affinity to ribosomes and it is unclear if DarB:Rel interaction is or is not comparable with ribosomal recruitment. I would guess that sequestration Rel into dimers by DarB – if this happens – would reduce the active concentration of

Rel in the cell significantly and compromise the ribosome-dependent activity. Therefore, it is crucial to bring the ribosomes in when discussing the potential Rel:DarB interaction / regulation module. In case performing the experiments with ribosomes is too challenging, please spell it out in Discussion that putting Rel:DarB interaction into the ribosomal context is essential and is an obvious direction for future research.

Rel protein preparation: line 381: 'as described previously 21.'. The authors have significantly modified the protocol for Rel preparation as compared to in paper cited. Please specify the parts that were altered. This is important given the significant difference in activity of your preparations, i.e. 10-fold lower HD activity.

MS data (supplemental Table 3): in EB using empty vector, 0.1 mM KCl (LK26) the authors have detected Rel (coverage: 8%, Number of peptide: 9). Similarly, Rel was detected in EB of DarB-Strep, 0.1 mM KCl (LK28): Rel was detected (coverage: 20%, Number of peptide: 57). While is 6.3 fold higher in DarB-Strep sample, but would Rel be detected by WB in EB in empty vector? Note that in EB of DarB-Strep, 0.5 mM KCl (LK36) Rel was not detected, while His6-tagged Rel is detectable on Figure 1c. While neither of these are crucial points, I the authors should be more careful when stating that the MS data clearly support idea that DarB binds to Rel only in potassium-depleted condition. Maybe one could describe quantitative data in SEC-MALS (supplemental Figure 3) in more detail: mere comparison in CBB-stained gel is not 100% convincing since the A280nm peak of DarBA25G, R132M looks similar in all conditions.

Line 14: 'recent study demonstrated the transient accumulation of (p)ppGpp upon heat stress 43'. Note that in E. coli activation of the stringent response upon heat shock was discovered by Gallant and colleagues in 1977 (Gallant et al., 1977), i.e. almost half a century ago.

Typos / minor corrections:

Line 78: "Rel" instead of RelA

Line 165: "Rel" instead of RelA

Line 379: "10xHis-SUMO-Rel" instead of 10xHis-SUMO-RelA.

Figure 4ab: please do check the labels: Rel, not RelA. Also, please spell out that DarBcdA is DarB+c-di-AMP. Please mention the concentrations in the legend.

References:

Benoist, C., Guerin, C., Noirot, P., and Dervyn, E. (2015). Constitutive Stringent Response Restores Viability of *Bacillus subtilis* Lacking Structural Maintenance of Chromosome Protein. *PLoS One* 10, e0142308.

Gaca, A.O., Kajfasz, J.K., Miller, J.H., Liu, K., Wang, J.D., Abranches, J., and Lemos, J.A. (2013). Basal levels of (p)ppGpp in *Enterococcus faecalis*: the magic beyond the stringent response. *mBio* 4, e00646-00613.

Gallant, J., Palmer, L., and Pao, C.C. (1977). Anomalous synthesis of ppGpp in growing cells. *Cell* 11, 181-185.

Krasny, L., and Gourse, R.L. (2004). An alternative strategy for bacterial ribosome synthesis: *Bacillus subtilis* rRNA transcription regulation. *EMBO J* 23, 4473-4483.

Kriel, A., Bittner, A.N., Kim, S.H., Liu, K., Tehranchi, A.K., Zou, W.Y., Rendon, S., Chen, R., Tu, B.P., and Wang, J.D. (2012). Direct regulation of GTP homeostasis by (p)ppGpp: a critical component of viability and stress resistance. *Mol Cell* 48, 231-241.

Lemos, J.A., Lin, V.K., Nascimento, M.M., Abranches, J., and Burne, R.A. (2007). Three gene products govern (p)ppGpp production by *Streptococcus mutans*. *Mol Microbiol* 65, 1568-1581.

Potrykus, K., Murphy, H., Philippe, N., and Cashel, M. (2011). ppGpp is the major source of growth rate control in *E. coli*. *Environ Microbiol* 13, 563-575.

Takada, H., Roghanian, M., Murina, V., Dzhygyr, I., Murayama, R., Akanuma, G., Atkinson, G.C., Garcia-Pino, A., and Haurlyuk, V. (2020). The C-Terminal RRM/ACT Domain Is Crucial for Fine-Tuning the Activation of 'Long' RelA-SpoT Homolog Enzymes by Ribosomal Complexes. *Front Microbiol* 11, 277.

Varik, V., Oliveira, S.R.A., Haurlyuk, V., and Tenson, T. (2017). HPLC-based quantification of bacterial housekeeping nucleotides and alarmone messengers ppGpp and pppGpp. *Sci Rep* 7, 11022.

Reviewer #2 (Remarks to the Author):

In their revision, the authors have satisfactorily addressed my concerns and the manuscript has been significantly strengthened.

Reviewer #3 (Remarks to the Author):

The revised manuscript looks much improved compared to the original one. The revised version has included almost all aspects that were lacking in the first version of the manuscript. It is a comprehensive piece of work on the interaction between c-di-AMP binding protein DarB and stringent response protein Rel. Researchers investigated these interactions using multiple methods to validate the interesting hypothesis. However, there are still following minor issues that should be resolved before publication;

1. In the method section of ITC, the concentration of RelNTD is mentioned as 5 μ M while in figure legends of ITC graph (2e) it is mentioned as 10 μ m. Please make it consistent with the methods. It will be more appropriate if the authors mention Kd value and stoichiometry in the inset of the processed graph (figure 2e).

2. The figure legends of figure 2a, description of the RRM is as ribosomal recognition motif. Actually, it is RNA Recognition Motif (RRM). It will be better if the authors write their alternate name as ACT domain as described in many publications.

3. Since the authors have corrected the name of the protein as Rel however some places it is left as RelA (headings and figure-4) therefore it should be corrected as Rel to make consistency with the nomenclature of protein.

4. The manuscript text has several spelling typos that need to be corrected such as pg 2 line 44 "homeostasis", pg 8 line 167, "terminal", pg 9 line 204 "concentration" pg 11 line 240 "stimulates" pg 19 line 452 "length".

Response to the reviewers

Reviewer #1

As I have pointed out in the previous review, the authors convincingly argue that DarB can bind the NTD domain region of Rel, but both the mechanistic implications (i.e. what happens with (p)ppGpp synthesis and degradation by Rel) and the physiological significance of this interaction are not so clear. In the revised version, Krüger and colleagues did a great job at significantly beefing up the manuscript and expanding the methodology they use. They have put a significant effort into generating new data which would enhance the mechanistic aspect of the study.

RE: We are grateful that the reviewer appreciates the amount of work that went into the revision, and we are also happy that he thinks that the revised manuscript has been improved.

I understand that the authors are under certain level of time pressure since the results were already presented at the Subtillery 2020 conference, and I would guess that several labs are already actively working to validate the results. That said, unfortunately, it feels that the authors have rushed the re-submission. There was no need: I am sure that there is no chance of scooping, given how small the *B. subtilis* stringent response community is.

I am absolutely positive that we all want to see this paper published so we can build on these results.

Since the effects reported in the revised version are modest or / and in direct conflict with a significant body of literature, an additional round of revision is, unfortunately, necessary.

Major concern: As Reviewer 3 has pointed out, Δ rel *B. subtilis* strain has a growth defect and requires amino acid supplements to offset this defect. The authors reply by saying that “rel mutant have a growth defect in complex medium but not in minimal medium. ... Under our conditions in minimal medium, essentially no (p)ppGpp is formed in the rel mutant which is in perfect agreement with the absence of a growth defect.”. This statement is very surprising and does not agree with the published data.

Several groups reported that in (p)ppGpp is essential for growth on minimal medium in the case of *E. coli* (Potrykus et al., 2011) and *B. subtilis* (Kriel et al., 2012). In the revised version, Krüger and colleagues report that there is no detectable pppGpp or ppGpp production in Δ rel mutant grown on minimal medium (Figure 4C). This is highly surprising: if the strain would have no (p)ppGpp, it would not grow on minimal medium at all. Δ rel *B. subtilis* should have (p)ppGpp levels much higher than the wild type: in the absence of Rel HD activity, (p)ppGpp synthesised by the SAS enzymes is not degraded, resulting in growth defect. One could argue that, somehow, in minimal media the situation is different. However, constative (p)ppGpp production was reported for *B. subtilis* Δ rel mutant grown on minimal medium in (Benoist et al., 2015). Similarly, both *S. mutants* (Lemos et al., 2007) and *E. faecalis* (Gaca et al., 2013) Δ rel mutants grow much slower in synthetic medium compared with wild type due to accumulation of SAS-generated (p)ppGpp. Similarly, Δ rel *B. subtilis* engineered to express Rel NTD (which is virtually enzymatically inactive) under IPTG-inducible promoter was tested in growth assays on minimum and rich medium in the absence of IPTG (Takada et al., 2020). Under these conditions the strain is basically Δ rel, with no NTD expressed. As expected – and in direct contradiction with the statement from Krüger and colleagues – the growth defect of Δ rel *B. subtilis* was considerably more pronounced on minimal medium.

RE: Please be sure that we knew the published papers as well as the reported phenotypes! Therefore, we were originally also quite sceptical about the validity of our findings. However,

we have repeated the experiments several times, and always got the same results! Personally, I was absolutely surprised to see no (p)ppGpp in the *rel* mutant, and as the reviewer, I did not want to believe it. But the same result was obtained both before and after the lockdown in many independent experiments. We also have checked the identity of the strains, so there is no doubt about the results.

However, we must say that the reviewer generalizes minimal medium: It is clear that very different media can be used and that the availability and concentration of potassium and glutamate are crucial for many physiological and regulatory responses. Indeed, we have tested growth of the *rel* mutants in minimal medium at potassium potassium and/or glutamate levels, and we observed, that there is not much difference between the wild type and the *rel* mutant if ammonium is used as the nitrogen source (the conditions relevant for all our experiments reported in the paper). In contrast, we see a severe growth defect of the *rel* mutant when glutamate was present as the nitrogen source! Thus, our results are valid under our experimental conditions. These conditions were chosen because they are relevant for the c-di-AMP levels that control DarB binding to Rel!

This is now shown in Supplementary Fig. 9, and we have also included a corresponding statement to the text (p. 9, highlighted). For your convenience, please see Fig. S. 9 below:

Supplementary Fig. 9. Phenotypes of a *B. subtilis rel* mutant. Growth assay of *B. subtilis* wild type, and GP3419 (Δrel). The *B. subtilis* strains were cultivated in MSSM minimal medium with 0.1 mM KCl and ammonium. The cells were harvested, washed, and the OD₆₀₀ was adjusted to 1.0. Serial dilutions were dropped onto MSSM minimal plates with the indicated potassium concentration and ammonium or glutamate, or on sporulation (SP) and LB complex medium plates.

As an addition, I would also like to mention that we have observed that one of the small alarmone synthetases of *B. subtilis*, *SasA*, forms ppGpp in response to the availability of glutamate. This unpublished observation supports the conclusion that potassium/ glutamate are important players in the control of (p)ppGpp synthesis and the corresponding responses. Again, the precise composition of the minimal medium is of key importance, and we must not take one minimal medium for all (as *E. coli* is not representative for the elephant!).

Essential experimental revision: Collectively, this suggests that something is seriously wrong with Δrel *B. subtilis* used by Krüger and colleagues. This is not all that surprising. In our hands this strain is

quite tricky to work because it picks up suppressor mutations (I would guess abrogating production of SAS), especially when grown on minimal medium. Since both HPLC and growth data do not fit the reported literature, I suggest requesting Δrel *B. subtilis* from some other lab (I recommend the strain from Yuzuru Tozawa, it is well-characterised). The issues outlined above affect Figures 3B and 4C and should be addressed.

RE: See above! Our strains have been checked and nothing is wrong with them! Our lab has more than 20 years' experience of genetic work with *B. subtilis*, and this includes very challenging projects such as the consecutive deletion of 40% of the *B. subtilis* genome (Reuß et al. 2017, Genome Research) or the investigation of essential and quasi-essential genes and functions (Gundlach et al. 2017, Sci. Signal.; Reuß et al. 2018, Nucleic Acids Research). Moreover, we have extensive experience with the isolation and characterization of suppressor mutants (just the PubMed IDs here, since these are many papers: 32743959, 32253343, 31061098, 30957856, 30181249, 29242163, 28679749, 28579978, 28420751, 28294562, 27260660, 26240071, 24939848, 23869754, 23192352, 22178973, 21219666, 18326565, 17183217). So we don't think this is a valid point!

Essential text revision: Please provide a Supplementary table listing the strains used in the paper. Since the strain uses several strains (some of which were constructed specifically for this paper, including the potentially problematic Δrel , see above), it is essential to document them.

RE: We use only three different *B. subtilis* strains (wild type, *darB* and *rel* mutants), and the construction of the mutants is described in **METHODS**. We therefore feel, that a Table is not necessary for only two strains.

Essential figure revision: In the revised version the authors report an HPLC analysis of nucleotide pools. These results crucial for assessing the physiological relevance of the Rel:DarB interaction. Since the authors use HPLC, they simultaneously measure several nucleotides (GTP / GDP; ATP / ADP; pppGpp / ppGpp). In the case of *B. subtilis*, the critical control parameter is not the concentration of (p)ppGpp; it is the ATP / GTP ratio (Krasny and Gourse, 2004). Since the authors already have the data, please plot the results as fractions of the nucleotide pools: the G pool, the A pool, the NTP pool; and please calculate the adenylate energy charge (AEC) so we can assess the quality of the data, it should be 0.8-0.95 (e.g. see (Varik et al., 2017)). No extra work is needed, all that is needed is re-plotting the data.

RE: Actually, we did not measure NTPs and NDPs, therefore we are unable to include these data.

Essential experimental revision: However, what is needed – in order to put the results into perspective – is a positive control, i.e. acute stringent response induced by SHX / mupirocin. Right now, the data are presented in pmox x OD600 X ml⁻¹ of ppGpp / pppGpp, and it is hard to assess if the 2-fold increase is meaningful. The levels of (p)ppGpp vary dramatically across the growth curve (at least 20-fold (Varik et al., 2017)), so a mere 2-fold change could simply reflect the difference in growth rate/position on the growth curve, and not a specific effect of DarB on Rel SYNTH / HD. Stress caused by expressing a mildly toxic protein such as DarB could easily cause a 2-fold increase in (p)ppGpp.

RE: We are surprised that the reviewer brings this point now and did not come up with it in the first round of evaluation. We have determined the (p)ppGpp levels always at the same

optical density, thus under comparable physiological conditions. A twofold effect of DarB on (p)ppGpp synthesis was also observed in *Listeria monocytogenes* (Peterson et al., 2020, <https://mbio.asm.org/content/11/4/e01625-20.long>). However, we have to take into account that we calculate the (p)ppGpp levels corresponding to the culture density. What we do not know, are the subcellular pools of the nucleotide. This is a key question in second messenger research, therefore even comparably weak effects determined at the cellular level may be caused by significant changes in relevant subcellular pools. Unfortunately, there is no way to determine these subcellular pools.

Moreover, we would not call DarB a “mildly toxic protein”. The toxic effect is indirect, and this is clearly demonstrated by the fact that deletion of *rel* in the strain overexpressing DarB suppresses the growth defect, i. e. toxicity. This means, not DarB is toxic, but the specific activation of Rel as shown by *in vivo* and *in vitro* experiments (and as confirmed by the competing group).

Essential experimental revision: Since the effects are so weak (about 2x increase in (p)ppGpp) it is crucial to demonstrate that the growth defect caused by DarB overexpression is due to (p)ppGpp accumulation (direct causation) and not the opposite (growth defect caused by DarB toxicity is the cause of 2-fold (p)ppGpp increase). To do so, the authors should co-overexpress a hydrolase (e.g. MESH1 or SYNTH-inactive Rel) with DarB. This should restore the growth rate to wt (i.e. if we deal with causation, as the model implies). I notice that the authors have already received some plasmids from the Haurlyuk lab. I am sure the necessary materials for this experiment will be shared as well.

RE: This was not mentioned in the first review. Our genetic data (deletion of *rel* in the *darB* overexpression strain) clearly shows that the growth defect results from (p)ppGpp accumulation.

Minor revisions, largely addressable by rewriting / correcting the text and / or figures:

Model Figure 5: the model could (and should) be improved. Panel b displays DarB:Rel complex in the absence of c-di-AMP. Please integrate it into panel a (I would place it horizontally, the panel a) and show how c-di-AMP binds to DarB, dissociates the complex, show what happens with SYNTH / HD activates of Rel etc.; please connect the molecular model with the physiology one.

RE: We agree and have modified the figure accordingly.

Biochemistry: the experimental procedures need to be described: what are the molar concentrations of Rel, DarB, c-di-AMP? What is the intracellular range of c-di-AMP concentrations (about 1-5 μ M, right? Please provide the references). A specificity control (just c-di-AMP + Rel, no DarB) is missing in both SYNTH and HD activity test; c-di-AMP could be affecting Rel directly. It would be nice to have a titration of c-di-AMP and DarB. Note that the HD activity of your Rel is at least 10-fold lower that reported by Takada and colleagues, while you are using their construct and following (but not precisely, please see comment below) their purification procedure. Since the HD / SYNTH ratio is a key parameter for Rel (off the ribosome the factor is predominantly HD), please do mention this slight difference.

RE: The concentrations are provided in the METHODS section (p. 20). The requested specificity control is now included (see Fig. 4 A, B), and explaining statement have been added to the text (p. 10, 11, highlighted in green).

Concerning the difference in Rel purification, there is the following statement in the text:
“For the purification of Rel, the column was washed with 8 column volumes of 4M NaCl to remove RNA prior to elution of the protein with 100 mM and 250 mM imidazole.”

This resulted in a better separation of Rel from RNA. This procedure gave at least as pure Rel protein as compared to Takada et al, and this might explain the slight difference in the assay. Moreover, we have described our purification procedure.

Please add the quantitative data to the figures, e.g. Kds on the ITC panels (Fig 2e, Kd is 0.65 μ M) and growth rates on Figure 3abc.

RE: For the ITC, the quantitative data have been added as an inset to Fig. 2e.. Growth rates and growth curves are actually the same data in two ways of presentation. Usually, all journals request to show a given set of data in just one way. This is why we decided to present the growth curves which give the reader a direct visual and thus more intuitive impression of the differences.

ITC: please discuss the abundance of Rel and DarB in the cell and contrast these with the Kd (0.65 μ M is not very low; and this is in the absence of c-di-AMP; there is always some in the cell).

RE: We agree, and have added the following statement to the discussion (p. 13, highlighted):
“The physiological relevance of the interaction is not only supported by the Rel-dependent growth inhibition upon overexpression of DarB, but also by the intracellular concentrations of both proteins (0.6 and 0.9 μ M for Rel and DarB, respectively ⁴⁰) which are in the range of the K_D determined for their interaction.”

In the rebuttal the authors state: ‘For the biochemical assays, we did not add ribosomes to be able to see the clear effect of DarB.’ This is not a good argument. In the cell Rel is activated by the ribosomes even under normal growth: this is why deltaRRM mutant is toxic in a ribosome-dependant way under normal growth conditions. Rel has considerable affinity to ribosomes and it is unclear if DarB:Rel interaction is or is not comparable with ribosomal recruitment. I would guess that sequestration Rel into dimers by DarB – if this happens – would reduce the active concentration of Rel in the cell significantly and compromise the ribosome-dependent activity. Therefore, it is crucial to bring the ribosomes in when discussing the potential Rel:DarB interaction / regulation module. In case performing the experiments with ribosomes is too challenging, please spell it out in Discussion that putting Rel:DarB interaction into the ribosomal context is essential and is an obvious direction for future research.

RE: We have added a corresponding statement to the very end of the discussion (p. 14, highlighted):

“It will be interesting to study whether yet additional factors may control Rel activity to trigger the stringent response under specific stress conditions and whether and how interaction with the ribosome affects the outcome of the Rel-DarB interaction.”

Rel protein preparation: line 381: ‘as described previously 21.’. The authors have significantly modified the protocol for Rel preparation as compared to in paper cited. Please specify the parts that were altered. This is important given the significant difference in activity of your preparations, i.e. 10-fold lower HD activity.

RE: We do now describe our purification procedure, and refer only to solutions from ref. 21.

MS data (supplemental Table 3): in EB using empty vector, 0.1 mM KCl (LK26) the authors have detected Rel (coverage: 8%, Number of peptide: 9). Similarly, Rel was detected in EB of DarB-Strep, 0.1 mM KCl (LK28): Rel was detected (coverage: 20%, Number of peptide: 57). While is 6.3 fold higher in DarB-Strep sample, but would Rel be detected by WB in EB in empty vector? Note that in EB of DarB-Strep, 0.5 mM KCl (LK36) Rel was not detected, while His6-tagged Rel is detectable on Figure 1c. While neither of these are crucial points, I the authors should be more careful when stating that the MS data clearly support idea that DarB binds to Rel only in potassium-depleted condition. Maybe one could describe quantitative data in SEC-MALS (supplemental Figure 3) in more detail: mere comparison in CBB-stained gel is not 100% convincing since the A280nm peak of DarBA25G, R132M looks similar in all conditions.

RE: The identification of peptides by MS is much more sensitive than a Western blot, therefore one always has to expect to see proteins that are present in the cell lysate. We agree that a “clearly support” statement is not appropriate, and no such statement occurs in the revised version.

For the SEC-MALS experiment with DarBA25G, R132M no big difference between the two tested conditions (with and without c-di-AMP) can be expected since this mutant protein binds only weakly to Rel.

Line 14: ‘recent study demonstrated the transient accumulation of (p)ppGpp upon heat stress 43’. Note that in *E. coli* activation of the stringent response upon heat shock was discovered by Gallant and colleagues in 1977 (Gallant et al., 1977), i.e. almost half a century ago.

RE: We agree and have specified that our statement applies to *B. subtilis*.

Typos / minor corrections:

Line 78: “Rel” instead of RelA

Line 165: “Rel” instead of RelA

Line 379: “10xHis-SUMO-Rel” instead of 10xHis-SUMO-RelA.

Figure 4ab: please do check the labels: Rel, not RelA. Also, please spell out that DarBcdA is DarB+c-di-AMP. Please mention the concentrations in the legend.

RE: We have corrected all of them. DarBcdA is now also explained in the legend, and concentrations are mentioned as requested (highlighted).

Reviewer #2 (Remarks to the Author):

In their revision, the authors have satisfactorily addressed my concerns and the manuscript has been significantly strengthened.

RE: We are grateful to this reviewer.

Reviewer #3 (Remarks to the Author):

The revised manuscript looks much improved compared to the original one. The revised version has included almost all aspects that were lacking in the first version of the manuscript. It is a comprehensive piece of work on the interaction between c-di-AMP binding protein DarB and

stringent response protein Rel. Researchers investigated these interactions using multiple methods to validate the interesting hypothesis.

RE: We are grateful to this reviewer for the kind words about our work.

However, there are still following minor issues that should be resolved before publication;

1. In the method section of ITC, the concentration of Rel^{NTD} is mentioned as 5 μ M while in figure legends of ITC graph (2e) it is mentioned as 10 μ m. Please make it consistent with the methods. It will be more appropriate if the authors mention K_d value and stoichiometry in the inset of the processed graph (figure 2e).

RE: We are grateful for this remark and we corrected the concentration of Rel^{NTD} in the method section. We also added the K_d value and the stoichiometry in the graph.

2. The figure legends of figure 2a, description of the RRM is as ribosomal recognition motif. Actually, it is RNA Recognition Motif (RRM). It will be better if the authors write their alternate name as ACT domain as described in many publications.

RE: Actually, we replaced ACT by RRM domain as used by Takada et al. (2020). The term RRM domain is also widely used, the domain is annotated in SMART (http://smart.embl.de/smart/do_annotation.pl?DOMAIN=SM00360) and all other major databases.

We feel that RRM is preferable as compared to ACT since it is directly linked to function (RNA binding), whereas ACT stands only for a rather meaningless collection of proteins that were initially found to contain this domain (Aspartokinase, Chorismate mutase and TyrA). Moreover, we corrected the error in the legend of Fig. 2a.

3. Since the authors have corrected the name of the protein as Rel however some places it is left as RelA (headings and figure-4) therefore it should be corrected as Rel to make consistency with the nomenclature of protein.

RE: This has been corrected throughout!

4. The manuscript text has several spelling typos that need to be corrected such as pg 2 line 44 "homeostasis", pg 8 line 167, "terminal", pg 9 line 204 "concentration" pg 11 line 240 "stimulates" pg 19 line 452 "length".

RE: Thank you very much! All typos have been corrected!

REVIEWER COMMENTS

Reviewer #1 (Remarks to the Author):

While revising the manuscript, Krüger and colleagues decided to argue all of the experimental criticisms instead of addressing them directly. I find it very unfortunate. All of the referees, including myself, are keen on seeing this work published. However, it is important to perform the essential controls. The Stülke lab is one of the leaders in *B. subtilis* genetics and constructing and testing a couple of strains hardly presents a challenge. I am unsure as to why the authors are so reluctant, especially given the overall positive reviews. Therefore, I have to re-iterate my concerns and strongly suggest addressing them experimentally.

1. Krüger and colleagues dismiss my concerns about their rel KO strain being a suppressor. The fact that rel KO picks up suppressors is well-established, and location of these in SAS genes (*ywaC* and *yjbM*) was demonstrated by Srivatsan and colleagues in PLOS Genetics in 2008; see Fig 5 and 6. The suppressor strain is more or less ppGpp⁰, just like the relKO strain obtained by Krüger and colleagues (normally rel KO has abnormally high (p)ppGpp levels). The only way to detect these suppressors is by performing whole genome sequencing. This was not done (see Construction of mutant strains by allelic replacement, line 360). The serial dilutions provided in the revised version do not look right either: while dilutions 10⁻⁰ – 10⁻² clearly go down in number of colonies (note clear heterogeneity in the colony size in rel KO), something strange happens in 10⁻³. Therefore, I have to re-iterate my concerns: the authors seem to pick up textbook suppressors of rel KO. Please have a look at my previous round of comments re suggested experiments and address them directly instead of arguing. Please demonstrate with HPLC that your rel KO strain has overproduced (p)ppGpp when grown on some common media as it is expected for rel KO strain (and is not suppressor with compromised *ywaC* and *yjbM*). Addressing these concerns experimentally is essential.

2. Establishing the causality between DarB expression, (p)ppGpp levels and growth defect is essential. I have not picked up on the problem initially because the relA KO issues were not evident, this is why this was not brought up during the first round of revision; now they are. Overcoming the growth defect caused by DarB by co-expression of an SAH degrading (p)ppGpp (MESH1?) is absolutely essential given the very likely problems with rel KO. Since the manuscript by Krüger and colleagues is now freely available on bioRxiv and the authors are not keen on performing additional control experiments, we have performed some in my lab. We confirm the effects of DarB and c-di-AMP on Rel enzyme in biochemical assays, and I am now fully confident in this part of the paper. However, the effects of DarB and c-di-AMP are gone when we add ribosomes, the key regulator of Rel. Obviously, ribosomes are always present in the cell... Therefore, it is absolutely crucial to perform one simple experiment and check that overexpression of RSH HD that degrades (p)ppGpp suppresses the growth defect they observe upon overexpression of DarB, and that the growth defect, is, indeed, due to activation of Rel SYNTH and inhibition of Rel HD. This is a trivial but crucial

experiment. The Stülke lab has tremendous expertise in *B. subtilis* genetics and constructing one strain is trivial.

3. Presentation of the HPLC data: in the rebuttal the authors say 'Actually, we did not measure NTPs and NDPs, therefore we are unable to include these data.' This is worrisome. HPLC, by default, resolves the full set of cellular nucleotide species. Without measuring several species simultaneously, it becomes hard to normalise the data for ppGpp. Please provide your raw HPLC data as SI, because now I am not sure how reliable are your HPLC-based measurements, it could just be a question of extraction efficiency. Making sure that your AEC is high and NTPs are quantified to calculate the GTP vs GTP ratio is really crucial when you work with *B. subtilis*. My lab has modest expertise in *B. subtilis* genetics, but when it comes to analysis of nucleotide pools by HPLC I am quite confident in our expertise... these are essential steps.

Reviewer #2 (Remarks to the Author):

I have been asked to comment on the issues previously raised by Reviewer #1.

The (p)ppGpp field is complicated and contradicting results unfortunately are not uncommon. It appears that there is a discrepancy of how the Drel strain of the Stülke lab and the Drel strains of other labs grow in minimal medium. Since minimal media can be different and given the vast experience of the distinguished Stülke lab, I have no reason to disbelieve Stülke's results. To resolve the standoff, I'd suggest that the Stülke lab requests a Δ rel *B. subtilis* strain from another lab and do a simple growth curve in their minimal medium and compare it to their own Δ rel strain.

It is not a lot of work and if Stülke's strain passes the test, I'd support publication of Stülke's manuscript.

Response to the reviewer

Reviewer #2 (Remarks to the Author):

I have been asked to comment on the issues previously raised by Reviewer #1.

The (p)ppGpp field is complicated and contradicting results unfortunately are not uncommon. It appears that there is a discrepancy of how the Drel strain of the Stülke lab and the Drel strains of other labs grow in minimal medium. Since minimal media can be different and given the vast experience of the distinguished Stülke lab, I have no reason to disbelieve Stülke's results. To resolve the standoff, I'd suggest that the Stülke lab requests a Δ rel B. subtilis strain from another lab and do a simple growth curve in their minimal medium and compare it to their own Δ rel strain.

It is not a lot of work and if Stülke's strain passes the test, I'd support publication of Stülke's manuscript.

Response:

We requested rel mutants from three different labs and from the Bacillus Genetic Stock Center, an internationally recognized Center for the curation of Bacillus materials (strains and plasmids).

Of the three labs, we got mutants from the Gert Bange lab (see a link for relevant publications here: <https://pubmed.ncbi.nlm.nih.gov/?term=bange+stringent&sort=date>) and from the Michael Hecker lab (publications: <https://pubmed.ncbi.nlm.nih.gov/?term=hecker+michael+stringent&sort=date>).

Moreover, Vasili Hauryliuk promised to send a mutant but never did so far. From the Bacillus Genetic Stock Center, we got two strains from the large Bacillus mutant collection (Koo et al., 2017, Cell Syst., PubMed 28189581).

For each of the strains as well as our own mutant, we first re-sequenced the rel as well as the small ppGpp synthetase sasA and sasB alleles since rel mutants tend to acquire mutations in sasA and sasB. As already verified when we initially characterized our mutant, it did not contain any mutation in neither sasA nor sasB. The same was true for the strain from the Bange lab (BHS126) and one of the strains from the Koo collection (BKK27600). In contrast, the second strain from the Koo collection (BKE27600) carried a frame shift mutation in sasA. Finally, the strain from the Hecker lab was a rel point mutant, and had the synthetase domain of Rel inactivated, but the hydrolase domain was wild type, as were sasA and sasB. Since the concern about the rel mutant resulted from the absence of the hydrolase activity, we felt that this strain was not a suitable control. Similarly, we did not include the BKE27600 strain in the further analyses as it had a sasA suppressor mutation.

Second, we compared the growth of our rel mutant and the strains BHS126 and BKK27600 (Bange lab and Koo collection, respectively) in the minimal medium that was also used for the experiment in the paper and in complex (LB) medium. As you can see in the new Supplementary Figure (S11 Fig.), all

three rel mutants grew comparably in the minimal medium. Compared to the wild type, growth was slightly reduced for all three strains. In LB, again all three rel mutants gave the same result, but here, the growth defect as compared to the wild type was much more pronounced (as expected and as reported in the literature). Similarly, S9 Fig. of the manuscript shows the reduced growth of our rel mutant as compared to the wild type if glutamate was included in the minimal medium. Again, this inhibition of the rel mutant in such a medium is well documented in the literature.

We feel, that this information should be sufficient to resolve the whole issue.

REVIEWERS' COMMENTS

Reviewer #2 (Remarks to the Author):

The authors accomplished the test that I requested, with independent strains from two different labs, and the two strains passed the test successfully.

Therefore, I support that the manuscript be accepted as soon as possible.

RESPONSE TO REVIEWERS' COMMENTS

Reviewer #2 (Remarks to the Author):

The authors accomplished the test that I requested, with independent strains from two different labs, and the two strains passed the test successfully.

Therefore, I support that the manuscript be accepted as soon as possible.

RE: We are grateful to the reviewer.